# Tuning of olfactory cortex ventral tenia tecta neurons to distinct task elements of goal-directed behavior

Kazuki Shiotani[1,2†], Yuta Tanisumi[1,2†], Koshi Murata[1,3], Junya Hirokawa[1], Yoshio Sakurai[1], Hiroyuki Manabe[1*]

[1]Laboratory of Neural Information, Graduate School of Brain Science, Doshisha University, Kyoto, Japan; [2]Research Fellow of the Japan Society for the Promotion of Science, Tokyo, Japan; [3]Division of Brain Structure and Function, Faculty of Medical Sciences, University of Fukui, Fukui, Japan

**Abstract** The ventral tenia tecta (vTT) is a component of the olfactory cortex and receives both bottom-up odor signals and top-down signals. However, the roles of the vTT in odor-coding and integration of inputs are poorly understood. Here, we investigated the involvement of the vTT in these processes by recording the activity from individual vTT neurons during the performance of learned odor-guided reward-directed tasks in mice. We report that individual vTT cells are highly tuned to a specific behavioral epoch of learned tasks, whereby the duration of increased firing correlated with the temporal length of the behavioral epoch. The peak time for increased firing among recorded vTT cells encompassed almost the entire temporal window of the tasks. Collectively, our results indicate that vTT cells are selectively activated during a specific behavioral context and that the function of the vTT changes dynamically in a context-dependent manner during goal-directed behaviors.

*For correspondence:
hmanabe@mail.doshisha.ac.jp

†These authors contributed equally to this work

Competing interests: The authors declare that no competing interests exist.

## Introduction

In mammals, odor signals detected by sensory neurons in the olfactory epithelium are transmitted via the olfactory bulb to the olfactory cortex. Mitral and tufted cells in the olfactory bulb send axonal projections to various areas of the olfactory cortex (*Igarashi et al., 2012*; *Neville and Haberly, 2004*). Despite growing knowledge on how odor molecules are coded by olfactory sensory neurons (*Buck and Axel, 1991*) and how neural circuits in the olfactory bulb and olfactory cortex process odor signals (*Mori and Sakano, 2011*; *Mori et al., 2013*; *Wilson and Sullivan, 2011*), our understanding of how distinct regions of the olfactory cortex transform olfactory sensory information into behavioral outputs remains limited (*Choi et al., 2011*). In the present study, we examined how the ventral tenia tecta (vTT), a relatively unexplored area of the olfactory cortex located in the ventromedial aspect of the olfactory peduncle, transforms the perception of odor signals into reward-directed behaviors.

Physiological studies of the visual, auditory, and somatosensory cortices have revealed that neocortical sensory area neurons receive sensory signals from the environment, as well as top-down signals that are generated internally by higher order cognitive and behavioral decision processes, including attention, reward-prediction, working memory, and behavioral choice processes (*Allen et al., 2019*; *Gilbert and Sigman, 2007*; *Roelfsema and de Lange, 2016*). In olfactory circuits, olfactory tubercle neurons encode goal-directed behaviors and exhibit enhanced odor responses when animals selectively attend to specific odors (*Carlson et al., 2018*; *Gadziola and Wesson, 2016*). Furthermore, c-Fos expression (a marker of neuronal activity) increases in different cells in the olfactory tubercle during distinct motivated behaviors in mice (*Murata et al., 2015*).

However, the physiological and behavioral functions of the vTT are poorly understood. To examine whether vTT neurons are modulated by cognitive and behavioral decision-making processes, we recorded the spike activity of individual vTT cells during the performance of a series of odor-guided, goal-directed behaviors in mice.

We observed that the firing of individual vTT neurons during odor-guided goal-directed behaviors was highly tuned to distinct task-elements that occurred in relation to the flow of goal-directed tasks, with each task-element occurring in a specific behavioral context. Our results indicate that vTT functions are dynamic rather than fixed, whereby changes in information processing mode occur in a context-dependent manner during a sequence of feeding and drinking behaviors.

## Results

### Activity of vTT cells during the odor presentation phase of the odor-guided go/no-go task

To examine how vTT cell firing changes in relation to odor-guided, goal-directed behaviors, we recorded the neural activity of vTT cells during an odor-guided go/no-go task with a water reward (*Figure 1A*). In this task, a light stimulus presented at the right odor port signaled to a water-restricted mouse to start the task and insert its nose into the odor port. At the moment of the nose poke, an odor cue was presented for 500 ms in the odor port. The mouse was required to keep its nose in the odor port during odor presentation to sniff the odor cue. After odor presentation, the light was turned off, and the mouse could withdraw its nose from the odor port. If the go odor cue (eugenol) was presented, the mouse was required to move to and poke its head into the left water port within a timeout period of 2 s to obtain a water reward (go trial). At the water port, the mouse was required to keep its head in the port for 300 ms to wait for water delivery. A drop of water (6 μL) was delivered 300 ms after the head poke. If the no-go odor cue (amyl acetate) was presented, the mouse was required to stay near the odor port for 2 s after the end of odor delivery without poking its head into the water port (no-go trial). After the mice were well trained, their behavioral accuracy remained above 80% throughout the session. The median duration of nose pokes after the onset of odor stimulation was 839 ms (interquartile range: 703–1003 ms) in the go trials and 738 ms (interquartile range: 621–946 ms) in the no-go trials (57 sessions from six mice).

We recorded the spiking activity of 270 vTT cells from six mice (*Tables 1* and *2*; recording positions are shown in *Figure 1B*) while they performed the go/no-go task. As the vTT receives direct inputs from mitral and tufted cells of the olfactory bulb, we first focused on whether vTT cells exhibited odor cue-responsive activity during odor presentation. We observed that a subset of vTT cells increased their firing rates during the odor presentation phase during both go and no-go trials (an example is shown in *Figure 1C*). To quantify the dependence of firing rate on the odor presentation phase, we calculated firing rate changes from baseline (pre-odor cue period, 1.2 to 1 s before the odor port entry) in sliding bins (width, 100 ms; step, 20 ms) using a receiver operating characteristic (ROC) analysis approach. We calculated the area under the ROC curve (auROC) at each time bin (spike data were aligned to the onset of odor valve opening). auROC values ranged from −1 to +1, with positive and negative values reflecting increased and decreased firing rates relative to baseline, respectively. We further determined auROC value significance using a permutation test (see Materials and methods).

Using the auROC approach, we defined the odor cue-responsive population (n = 68 cells, 25% of the recorded cells) as cells that significantly increased their firing rates from baseline for five consecutive bins (100 ms) during the odor presentation phase (500 ms after the odor valve opening) in correct go or no-go trials. Across all the cells, many exhibited significant increases in firing rate relative to both the go and no-go odor cue presentation phases (top color maps in *Figure 1D*, p<0.01, permutation test). No significant differences in the magnitude of these increases between the go and no-go odor cue phases at each time point were observed (bottom lines in *Figure 1D*, p>0.01, Welch's *t* test). Changes in firing rate in individual vTT cells exhibited similar time courses for go and no-go trials. We quantified this by calculating the correlation coefficients of response profiles between correct go trials and correct no-go trials for each cell (top lines in *Figure 1E*). This analysis revealed that the activity of vTT cells was strongly correlated between go and no-go odor cue presentation phases, whereas different cell pairs did not exhibit this correlation (bottom lines in

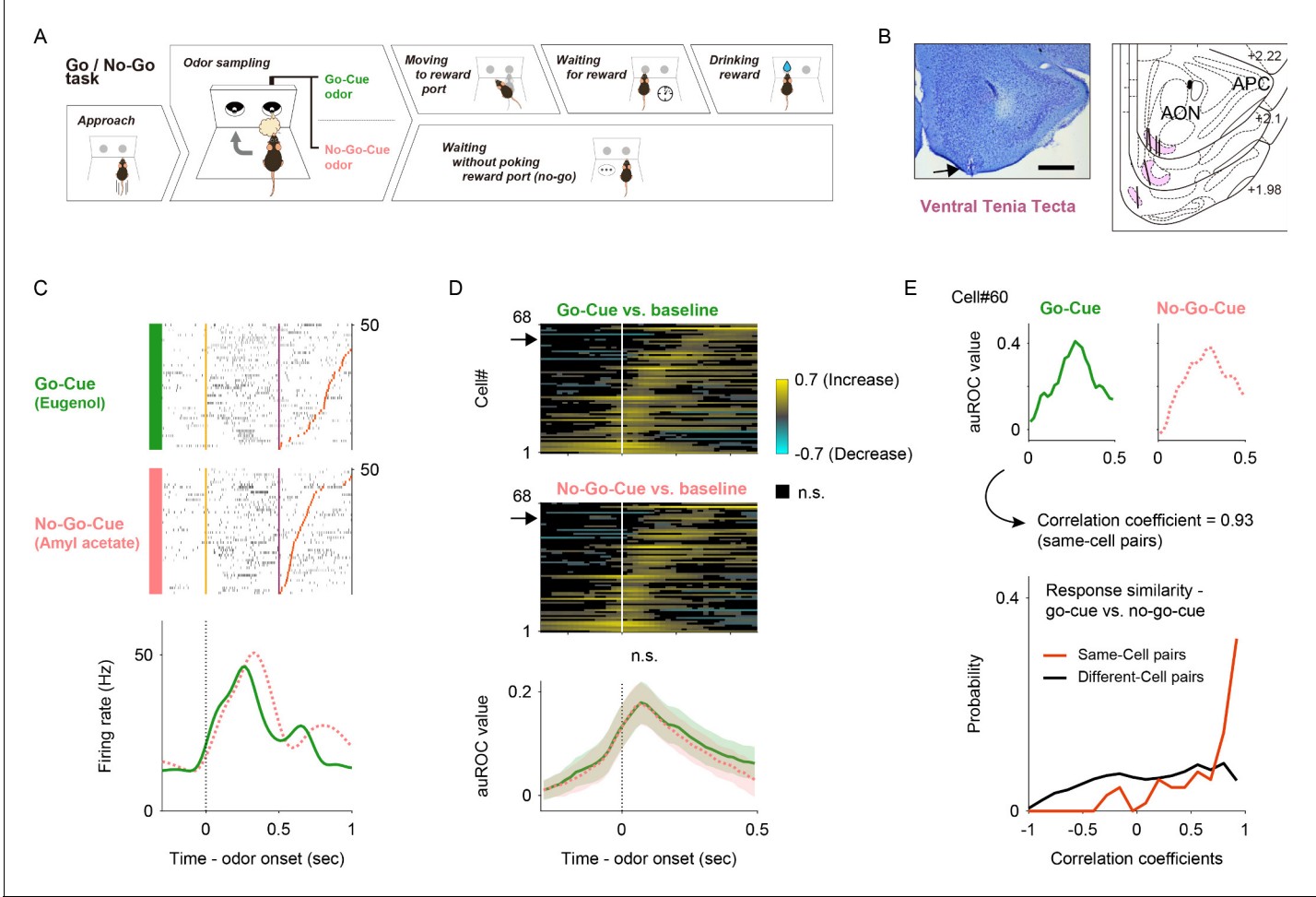

**Figure 1.** vTT cell activity patterns during the odor presentation phase of the odor-guided go/no-go task. (**A**) Schematic of the odor-guided go/no-go task. Behavioral epochs progressed temporally from left to right. (**B**) Nissl-stained frontal sections (arrow indicates recording track) and recording tracks (vertical thick lines) of the vTT in the odor-guided go/no-go task. The pink area shows layer II of the vTT. APC, anterior piriform cortex; AON, anterior olfactory nucleus. Scale bar: 500 μm. (**C**) Example firing patterns of vTT cells during the odor-guided go/no-go task. Each row contains spikes (black ticks) for one trial, aligned to the time of odor valve opening (corresponding to odor port entry, orange ticks). Purple and red ticks indicate times of odor valve closing and odor port exits, respectively. Correct trials are grouped by odor. Within each group, correct trials are sorted by the duration of the odor sampling epoch (the time from odor valve opening to odor port exit). Histograms are averaged across odors, calculated using a 20 ms bin width, and smoothed by convolving spike trains with a 60 ms-wide Gaussian filter. The vertical dashed line indicates the time of odor valve opening. (**D**) Upper panels: auROC values for cells with significantly higher firing rates than baseline (n = 68 cells) during the odor presentation phase (upper, go trials; lower, no-go trials). Each row corresponds to one cell, with cells in both graphs presented in the same order. auROC values (aligned by odor valve opening) were calculated by comparing go odor cue presentation and baseline (pre-cue odor period, 1.2 to 1 s before the odor port entry) or no-go odor cue presentation and baseline in sliding bins (width, 100 ms; step, 20 ms). Cells were sorted by peak value times of auROC values calculated by comparing go odor cue presentation and baseline for each cell. Vertical white lines indicate the time of odor valve opening. Color scale shows significant auROC values (p<0.01, permutation test). Black boxes indicate bins with non-significant auROC values (p>0.01, permutation test). Arrows indicate the same cell as that in (**C**). Lower panel: mean cell auROC values. Green line indicates go odor cue presentation versus baseline; pink dashed line indicates no-go odor cue presentation versus baseline. Shaded areas represent 95% confidence intervals. The vertical black dashed line indicates the time of odor valve opening. Note that across the population, mean auROC values did not differ significantly between go and no-go odor cue presentation at each time point (p<0.01, Welch's *t* test). (**E**) Neural response similarity measures between go and no-go odor cue presentation. Upper panel: examples of auROC values during the go/no-go odor cue presentation phase for one cell (same cell as in (**C**)) and the correlation coefficient of auROC values during the go/no-go-cue odor presentation phase. Lower panel: comparison of the correlation coefficients and cell-shuffled data, calculated for different pairs of cells. Neuronal response profiles were more similar between odor cues for paired responses from the same neuron (red) than for responses of two different neurons (black) (p<10$^{-13}$, two-sample Kolmogorov–Smirnov test).

**Table 1.** Basic information in the odor-guided go/no-go task.

| Mouse | Recording sessions | Trials/session | Go trials/session | No-Go trials/session | Recorded cells/session |
|---|---|---|---|---|---|
| #1 | 5 | 380 ± 31 | 190 ± 16 | 190 ± 15 | 3 ± 1 |
| #2 | 8 | 453 ± 27 | 227 ± 14 | 226 ± 13 | 6 ± 2 |
| #3 | 5 | 435 ± 44 | 217 ± 22 | 217 ± 22 | 4 ± 1 |
| #4 | 14 | 464 ± 21 | 233 ± 10 | 231 ± 10 | 7 ± 1 |
| #5 | 13 | 321 ± 24 | 161 ± 12 | 160 ± 12 | 3 ± 0 |
| #6 | 10 | 443 ± 24 | 222 ± 12 | 222 ± 12 | 5 ± 1 |

*Figure 1E*, $p < 10^{-13}$, two-sample Kolmogorov–Smirnov test). These results suggest that individual vTT cells did not represent odor cue differences between go and no-go trials during odor presentation phases. We therefore hypothesized that firing activity mainly reflected animal behavior and was dependent on task context.

## Behavior-specific activity of vTT cells in the odor-guided go/no-go task

Many vTT cells exhibited an increase in firing rate during specific behaviors over the course of the odor-guided go/no-go task (*Figure 2—figure supplement 1A*). Time intervals between behavioral events (the time from odor valve opening until the mouse withdrew its snout from the odor port, and the time from odor port withdrawal until reward port entry) also varied across trials (colored shaded areas in *Figure 2—figure supplement 1A*). To develop an overall firing profile accounting for this variability, we created event-aligned spike histograms (EASHs) (*Ito and Doya, 2015*). An EASH was derived by linearly scaling time intervals between behavioral events in each trial and the median interval for all trials (*Figure 2—figure supplement 1B*, see Materials and methods). The EASHs clearly demonstrated that individual vTT cells were activated during different behavioral epochs (between-event intervals), such as when mice were poking the odor port in the approach epoch (plots in bottom left, *Figure 2A*) and during the odor-sampling epoch (plots second from the bottom left, *Figure 2A*).

To quantify neural activity across behavioral epochs during the go trials, we calculated changes in event-aligned firing rates from baseline (pre-odor cue period, 1.2 to 1 s before odor port entry) in sliding bins (width, 100 ms; step, 20 ms) using ROC analyses. Across the population, almost all vTT cells exhibited a significant increase in firing rates during a specific behavior in each epoch (right color map in *Figure 2A*). Furthermore, individual vTT cells exhibited a significant decrease in firing rates in certain epochs. These results were independent of the size of the window in which auROC values were calculated (*Figure 2—figure supplement 2*). To evaluate the relationships between epoch-specific firing rates, vTT cells were classified into five groups based on their tuning peak time (*Figure 2—figure supplement 3*, see Materials and methods) with reference to the five go trial behavioral epochs. For each cell group, we calculated relative distributions of significant responses (*Figure 2—figure supplement 4*, see Materials and methods) for each bin, according to significantly increased or decreased changes in firing rate throughout the tasks (*Figure 2B*). Of vTT cells, 7% exhibited tuning peak times within the time window of the approach epoch, indicating a significantly

**Table 2.** The distribution of the vTT cell groups in the odor-guided go/no-go task.

| Mouse | Approach cells | Odor sampling cells | Moving cells | Waiting cells | Drinking cells | Others | (Total) |
|---|---|---|---|---|---|---|---|
| #1 | 0 | 10 | 1 | 0 | 2 | 0 | 13 |
| #2 | 6 | 8 | 13 | 7 | 10 | 4 | 48 |
| #3 | 0 | 5 | 7 | 1 | 2 | 3 | 18 |
| #4 | 6 | 21 | 15 | 11 | 18 | 32 | 103 |
| #5 | 3 | 8 | 16 | 5 | 1 | 7 | 40 |
| #6 | 4 | 14 | 11 | 5 | 7 | 7 | 48 |
| (Total) | 19 | 66 | 63 | 29 | 40 | 53 | 270 |

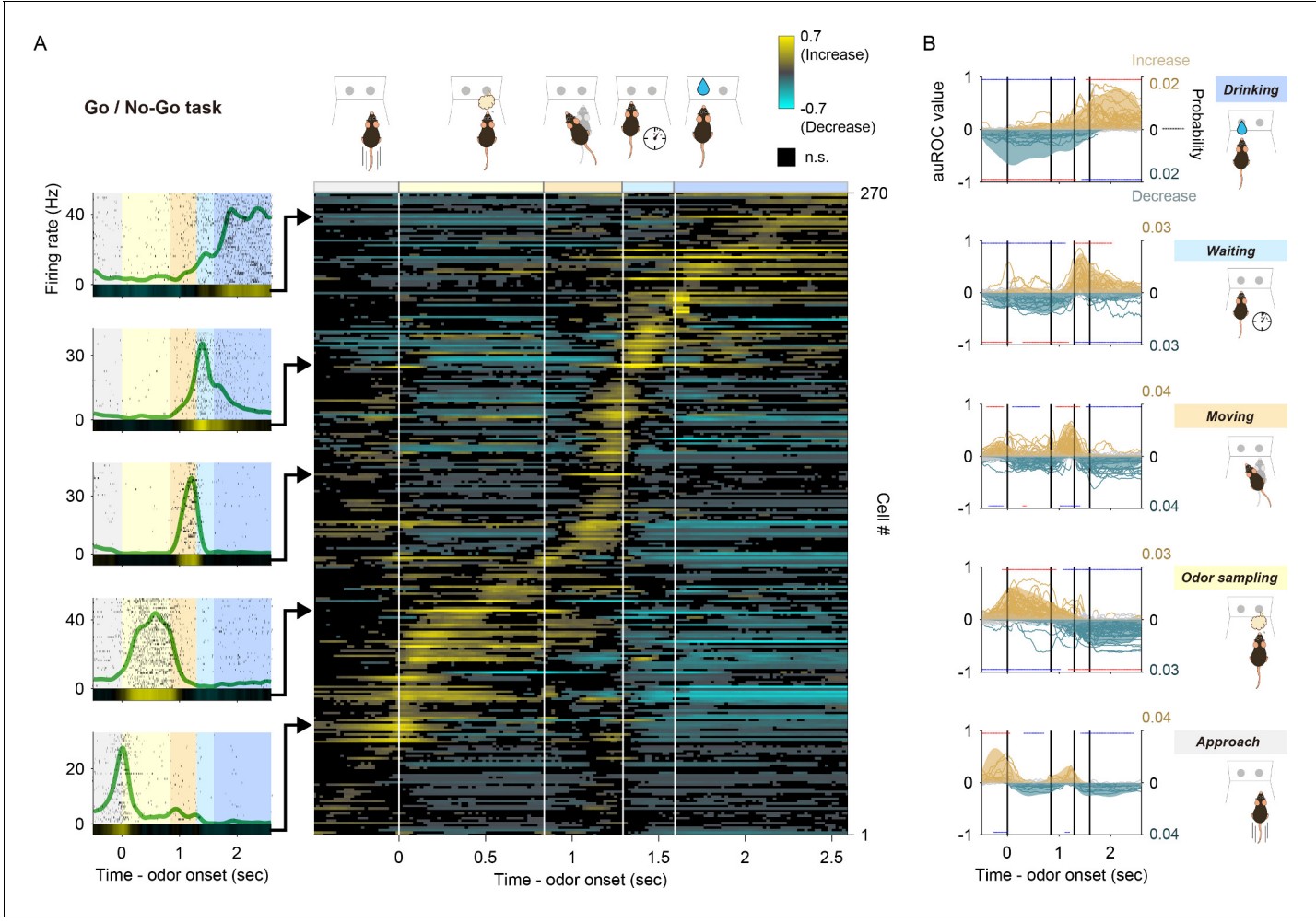

**Figure 2.** Tuning of vTT cells to distinct behaviors in the odor-guided go/no-go task. (A) Left panel: examples of event-aligned spike data for five representative cells tuned to specific behaviors. Event-aligned spike histograms were calculated using a 20 ms bin width and smoothed by convolving spike trains with a 60 ms wide Gaussian filter. Gray shading indicates the approach epoch (500 ms before odor port entry), yellow shading indicates the odor-sampling epoch (from entry into the odor port to exiting the odor port), orange shading indicates the moving epoch (from exiting the odor port to entry into the water port), light blue shading indicates the waiting epoch (water reward delay, 300 ms before water valve was turned on), blue shading indicates the drinking epoch (1000 ms after the water valve was turned on). Right panel: auROC values were calculated from event-aligned spike data (aligned by odor valve opening) for all cells, sorted by the peak time for auROC values. Each row corresponds to one cell. auROC values were calculated by comparing go correct trials to baseline (pre-odor cue period, 1.2 to 1 s before odor port entry) in sliding bins (width, 100 ms; step, 20 ms). Vertical white lines indicate transitions between behavioral epochs, including odor port entry (corresponding to odor valve opening), odor port exit, water port entry, and water valve opening. The color scale for this figure is the same as that used in *Figure 1D*, with positive and negative values reflecting increased and decreased firing rates relative to baseline, respectively. (B) Relative distribution of significant auROC values (p<0.01, permutation test) per cell group tuned to a specific behavioral epoch in the odor-guided go/no-go task. Cell groups were tuned to the approach, odor-sampling, moving, waiting, and drinking epochs (from bottom to top graphs). Each line corresponds to one cell (left axes, auROC values). Yellow and blue indicate a significant increase and decrease from baseline, respectively. Gray indicates neither a significant increase nor decrease. Shaded regions show the relative distributions of significant auROC valves (right axes, probability, see Materials and methods). Red dots indicate that a time bin contained more cells with significant responses than in the distribution of 1000 resampling datasets. Blue dots indicate that a time bin contained fewer cells with significant responses than in the distribution of 1000 resampling datasets. The resampling datasets are provided in *Figure 2—source data 1*. Vertical black lines indicate the timing of odor port entry (corresponding to odor valve opening), odor port exit, water port entry, and water valve opening. Note that each cell group exhibited an excitatory response to a specific behavioral epoch, with suppressed responses relative to other epochs.

The online version of this article includes the following source data and figure supplement(s) for figure 2:

**Source data 1.** Source data of the relative distributions of significant neural responses in odor-guided go/no-go task.
**Figure supplement 1.** Representative activity patterns of behavior-specific active vTT cells in the odor-guided go/no-go task.
**Figure supplement 2.** Dependence of auROC values on window size in the odor-guided go/no-go task.
**Figure supplement 3.** Tuning peak time and tuning duration.
*Figure 2 continued on next page*

*Figure 2 continued*

**Figure supplement 4.** Relative distribution of significant responses.

greater probability of exhibiting an increase in firing rate during the approach epoch; 24% had tuning peak times during the odor-sampling epoch, indicating that the firing rates of these cells were increased during the odor-sampling epoch; 23% and 11% had tuning peak times during the moving and waiting epochs, respectively, indicating that these cells were tuned to each epoch; and 15% had tuning peak times during the drinking epoch, indicating that they were tuned to the drinking epoch. Moreover, we observed that these classified cells tended to decrease their activity during other behavioral epochs, suggesting that individual vTT cells exhibit distinct tuning profiles for specific goal-directed behaviors.

## Behavior-specific activity of vTT cells in the odor-guided eating/no-eating task

We observed that the activity of vTT cells was modulated across behavioral epochs on a hundreds-of-milliseconds timescale (*Figure 2*). We subsequently investigated how vTT cells would respond during each epoch if the behavioral epochs were prolonged to a multi-second timescale. Moreover, we also assessed whether the activity pattern of vTT cells in the go/no-go task could be generalized to different contexts and motivated behaviors. To this end, we performed an odor-guided eating/no-eating task with food rewards (*Figure 3A*). Mice were placed on a food restriction schedule and trained to either choose to consume food in a dish or avoid consuming it using dish odor-cuing. We randomly presented sugar on a dish with one of three different odor cues (eugenol, vanilla essence, or almond essence). The dish was then placed at an arbitrary position in the test cage. During the learning phase, eugenol and vanilla essence odors were associated with a sugar reward, whereas the almond essence odor was associated with a sugar reward followed by aversive consequences (a LiCl injection). After the learning phase, mice would approach the dish and proceed to consume its contents if either the eugenol or vanilla essence odor was presented (eating trials). Approximately 6 s after the mouse began eating during the eating trials, the dish was removed even if the animal was still eating. The sequence of behaviors consistently exhibited by mice during the course of the eating trial epochs included approach (approaching the food dish), eating (from touching the dish to eating), and escape behaviors after food removal (*Figure 3A*). In contrast, mice would approach the dish but not consume its contents if almond essence odor was presented (e.g. during no-eating trials) (*Figure 3A*, No-eating). Powder (normal) chow was also presented on a dish in randomly interleaved trials. Behavior response accuracies for the cued odor trials and powder chow trials throughout all sessions were 0.93 ± 0.01, 0.90 ± 0.01, 0.96 ± 0.01, and 0.99 ± 0.003 (mean ± SEM, 63 sessions from six mice) for the eugenol, vanilla essence, almond essence, and powder chow trials, respectively.

We recorded spike activity from 374 well-isolated vTT cells from six mice (*Tables 3* and *4*, recording positions are depicted in *Figure 3—figure supplement 1*) while mice performed the odor-guided eating/no-eating task. We observed that most vTT cells demonstrated an increase in firing rates during a specific behavior during the odor-guided eating trials. The graphs in *Figure 3B* (left side) show examples of behavior-specific vTT cells with greatly increased firing rates during each behavioral epoch. The duration of mouse behaviors during the task (approach, eating, and other behaviors such as locomotion, grooming, and freezing during dish removal) varied between trials (*Figure 3—figure supplement 2A*). We generated EASHs for the behavioral epochs of the odor-guided eating/no-eating task to develop an overall firing profile accounting for behavioral timing variability (*Figure 3—figure supplement 2B*). The EASHs clearly revealed that individual vTT cells were activated at different behavioral epochs, such as during the touching of the dish in the approach epoch (plots in bottom left, *Figure 3B*) and eating epoch (plots second from the bottom left, *Figure 3B*).

To quantify the neural activity modulated by behavioral epochs, we calculated the changes in event-aligned firing rates from baseline (pre-approach period, 3 to 1 s before the onset of approach behavior) in sliding bins (width, 500 ms; step, 100 ms) using an ROC analysis approach. As shown in *Figure 3B*, most vTT cells exhibited an increase in firing in a specific behavioral epoch and a

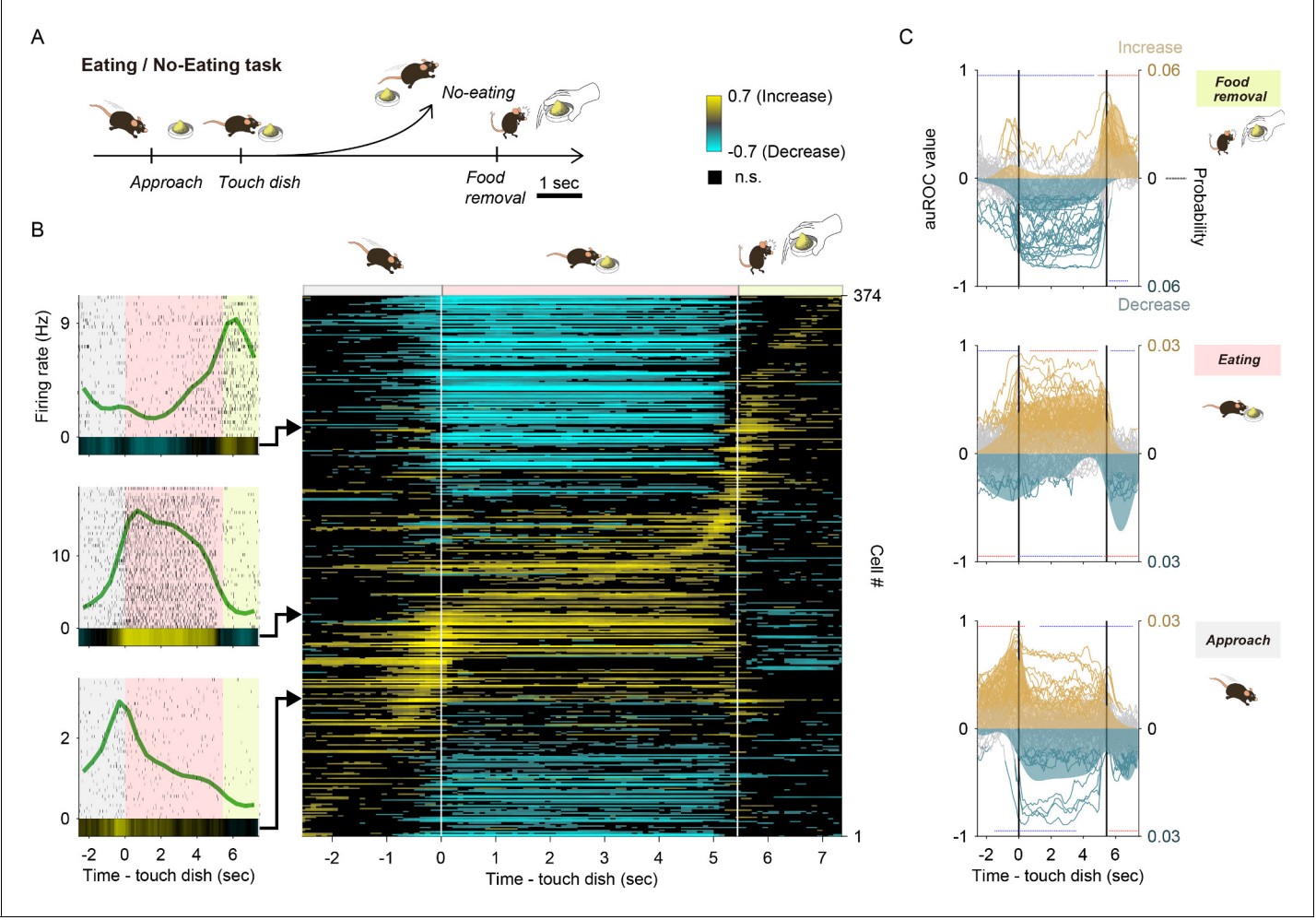

**Figure 3.** Tuning of vTT cells to distinct behaviors in the odor-guided eating/no-eating task. (**A**) Schematic of the odor-guided eating/no-eating task, with behavioral epochs progressing from left to right. (**B**) Left panel: examples of event-aligned spike data for three representative cells tuned to specific behaviors. The event-aligned spike histograms were calculated using a 500 ms bin width and smoothed by convolving spike trains with a 500 ms wide Gaussian filter. Gray shading indicates the approach epoch (from the start of approach behavior to contacting the dish), pink shading indicates the eating epoch (from the start of consumption to food removal), and light green shading indicates the food removal epoch (2 s after food removal). Right panel: auROC values calculated from event-aligned spike data (aligned to contacting the dish) for all cells, sorted by the timing of the peak of auROC values. Each row corresponds to one cell. auROC values were calculated by comparing eating trials versus baseline (pre-approach period, 3 to 1 s before starting approach behavior) in the sliding bins (width, 500 ms; step, 100 ms). Vertical white lines indicate when the animal touched the dish (left) and when the food was removed (right). The color scale indicates positive values that reflect increases in firing rate relative to baseline, while negative values reflect firing rate decreases relative to baseline. (**C**) Relative distribution of the significant auROC values (p<0.01, permutation test) in each cell group tuned to a specific behavior in the odor-guided eating/no-eating task. The cell groups were tuned to approach epoch, eating epoch, and food removal epoch (from bottom to top of graph). Each line corresponds to one cell (left axes, auROC values). Yellow and blue indicate a significant increase and decrease from baseline, respectively; gray indicates a non-significant increase or decrease. Shaded regions indicate the relative distribution of significant auROC valves (right axes, probability, see Materials and methods). Red dots indicate that a time bin contained more cells with significant responses than that in the distribution of 1000 resampling datasets. Blue dots indicate that a time bin contained fewer cells with significant responses than that in the distribution of 1000 resampling datasets. The resampling datasets are provided in the *Figure 3—source data 1*. Vertical black lines indicate when mice contacted the dish (left) and when food was removed (right). Note that each cell group had an excitatory response to a specific behavioral epoch, with suppressed responses relative to other epochs.

The online version of this article includes the following source data and figure supplement(s) for figure 3:

**Source data 1.** Source data of the relative distributions of significant neural responses in eating/no-eating task.
**Figure supplement 1.** vTT recording sites across mice in the odor-guided eating/no-eating task.
**Figure supplement 2.** Representative activity patterns of behavior-specific active vTT cells in the odor-guided eating/no-eating task.
**Figure supplement 3.** Dependence of auROC values on window size in the odor-guided eating/no-eating task.
**Figure supplement 4.** Eating epoch activity during eugenol, vanilla essence, and powder chow presentation in the odor-guided eating/no-eating task.

**Table 3.** Basic information in the odor-guided eating/no-eating task.

| Mouse | Recording sessions | Trials/session | Eating trials/session | No-Eating trials/session | Recorded cells/session |
|---|---|---|---|---|---|
| #1 | 6 | 48 ± 2 | 36 ± 2 | 12 ± 0 | 6 ± 2 |
| #2 | 10 | 44 ± 1 | 33 ± 1 | 11 ± 0 | 7 ± 1 |
| #3 | 6 | 55 ± 2 | 44 ± 2 | 11 ± 0 | 6 ± 1 |
| #4 | 12 | 59 ± 1 | 48 ± 1 | 12 ± 0 | 8 ± 1 |
| #5 | 10 | 55 ± 2 | 45 ± 2 | 11 ± 0 | 5 ± 1 |
| #6 | 18 | 58 ± 0 | 48 ± 0 | 11 ± 0 | 5 ± 1 |

decrease in firing in the other two behavioral epochs. These results were independent of the size of the window in which auROC values were calculated (*Figure 3—figure supplement 3*). To evaluate these distinct patterns, we classified individual vTT cells into three types based on their tuning peak time with reference to the three different eating trial behavioral epochs. These analyses revealed that 19%, 20% and 12% of vTT cells had tuning peak times during the approach, eating and food removal epochs, respectively, suggesting that these three cell classes were tuned to specific and distinct behaviors.

To evaluate the firing pattern of these cellular subtypes during the eating trials, we calculated the relative distributions of significant responses for each bin according to significantly increased or decreased changes in firing rate throughout the tasks (*Figure 3C*). For each cell group, the occurrence probability in each bin with significantly increased firing rates during the most preferred epoch (i.e. the epoch in which vTT cells had a peak tuning time) was significantly higher than that of the cell-shuffled data. The occurrence probability of significantly decreased firing rate bins tended to emerge in the other epochs. These results suggest that vTT cells were tuned to specific odor-guided eating behaviors on a multi-second timescale.

To address whether the increased firing of cells during the eating epoch was driven by the odors associated with the food dish, we compared mean cellular auROC values of the eating epoch during eugenol, vanilla essence, and powder chow presentation across cells with tuning peak times during the eating epoch (*Figure 3—figure supplement 3*). A significant difference in firing rate increases between vanilla essence and powder chow presentations was observed. No significant difference was observed in firing rate increases between eugenol and vanilla essence presentation or between eugenol and powder chow presentation, implying that vTT cells increase their firing during eating epoch behaviors in a manner largely independent of the type of odor cue.

## Comparison of behavior-specific activity of vTT cells between different odor-guided tasks

To further identify the common properties of vTT cell activity across distinct tasks, we first assessed the firing properties of vTT cells between tasks. No significant differences in baseline firing rates were observed between tasks (*Figure 4—figure supplement 1A*, p>0.05, Wilcoxon rank-sum test). No significant differences were noted in mean firing rates in the classified cell populations during each of the most preferred epochs (*Figure 4—figure supplement 1B*, p>0.05, Tukey's test). Analysis

**Table 4.** The distribution of the vTT cell groups in the odor-guided eating/no-eating task.

| Mouse | Approach cells | Eating cells | Removal cells | Others | (Total) |
|---|---|---|---|---|---|
| #1 | 9 | 13 | 0 | 13 | 35 |
| #2 | 31 | 20 | 1 | 17 | 69 |
| #3 | 4 | 7 | 4 | 19 | 34 |
| #4 | 7 | 24 | 9 | 51 | 91 |
| #5 | 8 | 4 | 15 | 21 | 48 |
| #6 | 12 | 8 | 16 | 61 | 97 |
| (Total) | 71 | 76 | 45 | 182 | 374 |

of the anatomical distribution of the classified cells in both tasks revealed a varied distribution pattern within the vTT (*Figure 4—figure supplements 2* and *3*). We then characterized and compared the firing time length and peak firing positions of vTT during the most preferred epoch. We investigated whether the tuning durations (*Figure 2—figure supplement 3*, see Materials and methods) of vTT cells corresponded to the actual durations of behavioral epochs. Three behavioral epochs (odor-sampling, moving, and waiting) were selected from the go/no-go task. The approach and eating epochs of the eating/no-eating task were also assessed based on the precisely defined start and end times of behavioral epochs within each trial. We then compared the distribution of tuning durations with behavioral durations in each epoch. The median for neural data was positively correlated with that of behavioral data (r = 1.00, p<10$^{-3}$, *Figure 4A*), suggesting that the duration of sustained discharge in vTT cells was dependent on the duration of the behavioral epoch. We then examined the distribution of tuning peak times in relation to the time course of tasks. We aligned the tuning peak times of vTT cells as a function of task time course (*Figure 4B*, top). We observed that the repertoire of vTT tuning peak times encompassed almost the entire continuing series of behavioral epochs for odor-guided reward-directed behaviors. Thus, tuning durations overlapped with all task epochs.

We also observed that the distribution of tuning peak times tended to increase around transitions to the next behavioral epoch, such as when animals explored the odor and water ports during the go/no-go task, or arrived or departed from the dish in the eating/no-eating task (*Figure 4B*, middle).

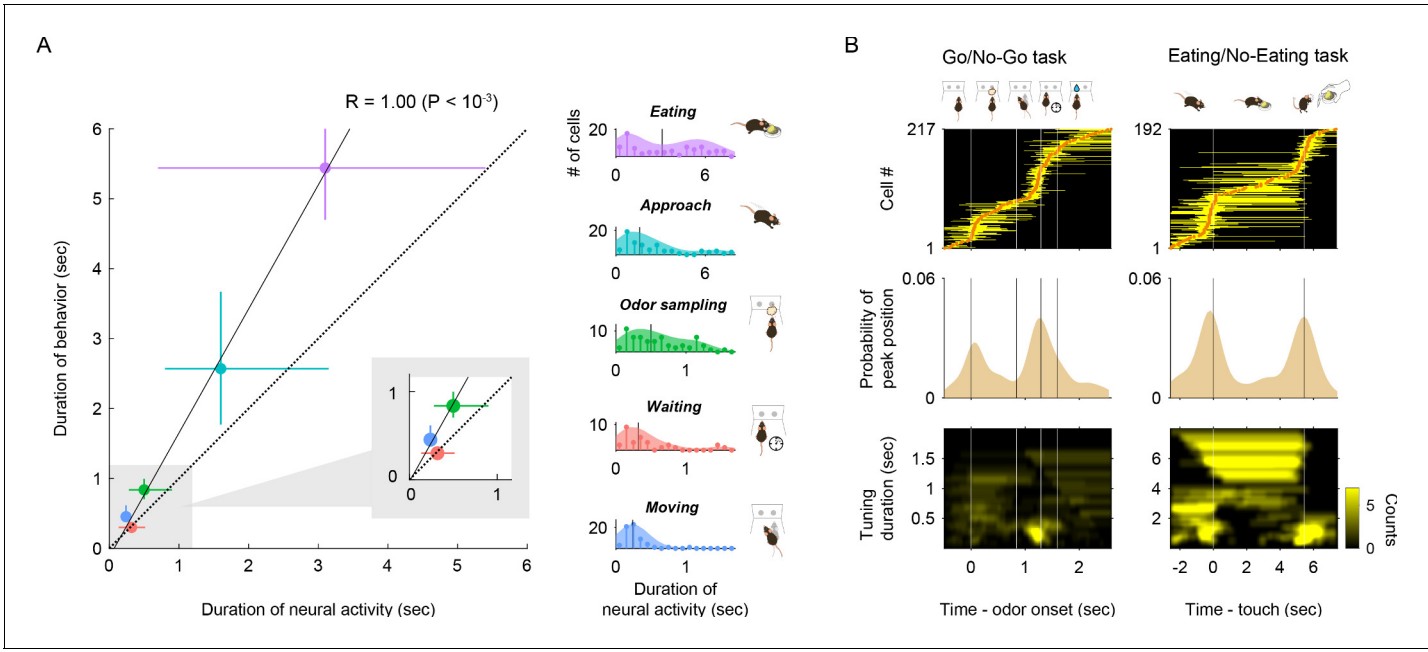

**Figure 4.** Tuning durations and tuning peak time of vTT cells in different tasks. (**A**) Correlation between the median of neural tuning durations and that of behavioral epoch durations (r = 1.00, p<10$^{-3}$, error bars show interquartile ranges). Right histograms show the distribution of neural tuning durations for each behavioral epoch (light blue, cell group tuned to approach epoch; purple, cell group tuned to eating epoch; green, odor sampling epoch; blue, moving epoch; red, waiting epoch). Vertical black lines in right graphs show the median tuning duration for each epoch. (**B**) Response profiles for tuning peak time and tuning duration across cells with tuning peak times (odor-guided go/no-go task, 217 cells; odor-guided eating/no-eating task, 192 cells). Top panel: tuning peak times (orange dots) and tuning durations (yellow horizontal lines) in the odor-guided go/no-go task and odor-guided eating/no-eating task. Vertical white lines indicate transitions between the behavioral epochs. Center panel: distributions of tuning peak times along the task time course. Shades show tuning peak time distributions. Vertical black lines show transitions between the behavioral epochs. Bottom panel: distributions of excitatory tuning durations, which were defined as the duration of consecutive significant bins (p<0.01, permutation test) with a 2-D Gaussian smoothing kernel. Vertical white lines indicate transitions between behavioral epochs. The color scale indicates the number of cells with each tuning duration in the tasks.

The online version of this article includes the following figure supplement(s) for figure 4:

**Figure supplement 1.** Comparison of baseline and mean firing rates of each classified group between tasks.

**Figure supplement 2.** Anatomical distribution of classified vTT cells in the odor-guided go/no-go task.

**Figure supplement 3.** Anatomical distribution of classified vTT cells in the odor-guided eating/no-eating task.

**Figure supplement 4.** vTT cells with tuning peak times in the eating epoch consisted of phasic and tonic responsive cells.

Furthermore, a subset of the cells with tuning peak times around transitions between behavioral epochs tended to have shorter tuning duration (*Figure 4B*, bottom), suggesting phasic activity around transitions. Indeed, during the eating epoch, the distribution of tuning durations exhibited a bimodal pattern (*Figure 4A*, *Figure 4—figure supplement 4*). Most phasic responses tended to be located at the end of the eating epochs. Collectively, these results suggest that individual vTT cells were tuned to specific time windows (both tuning durations and tuning peak times) during goal-directed behaviors.

## Behavioral context-dependent activity of vTT cells

We observed that many vTT cells tended to increase their firing rates before the most preferred behavioral epochs, and some cells maintained their increased firing rate after the epoch (*Figures 2B*, *3C* and *4B*). We therefore hypothesized that individual vTT cell firing was tuned to a specific behavioral context and was thus dependent on specific task-elements during task progression rather than on specific behaviors. If this were true, the response patterns of vTT cells would change in a different context, even for the same behavior. We assessed nose poke behaviors during the odor-sampling epoch of the odor-guided go/no-go task and observed that a subset of cells exhibited differences in firing rates between the go and no-go trials just before the animals exited the odor port, despite involving the same nose-poking behavior. We quantified this by calculating go-cue versus no-go-cue trial preference using ROC analyses with real-time data. When the differences in firing rate were aligned to odor port exit times, we observed that more cells had differences in firing rate between the go and no-go trials before the animal exited from the odor port compared to that during the odor presentation phase (*Figure 5A*). A subset of cells with tuning peak times not during the odor-sampling epoch also exhibited differences in firing rate just before odor port exit times.

To examine whether the population activity of vTT cells could account for behavioral accuracy, we performed a decoding analysis to determine whether the firing rates of vTT cell populations could be used to classify each individual trial as go or no-go. We used support vector machines with linear kernels as a decoder (*Cury and Uchida, 2010*; *Miura et al., 2012*). Analyses of the decoding time course based on all 270 cells using a sliding time window revealed that decoding accuracy was maintained at chance levels during odor presentations and subsequently increased close to the level achieved during behavioral accuracy just before odor port exit (*Figure 5B*). These results suggest that vTT cells were tuned to task elements in a particular behavioral context during goal-directed behaviors.

## Cell types and connectivity patterns of vTT cells

Although the majority of neurons in layer II of the vTT comprise pyramidal cells, the vTT also contains additional cell types (*Haberly and Price, 1978*; *Neville and Haberly, 2004*). To examine the distribution of glutamatergic and GABAergic cells in the vTT, we performed in situ hybridization to measure vesicular glutamate transporter 1 (VGluT1) and glutamic acid decarboxylase (GAD) 67 and GAD65 mRNA (*Slc17a7* and *Gad1/2* mRNAs, respectively) in the vTT (*Figure 6A,B*). Approximately 86% and 8% of vTT cells were *Slc17a7*-positive and *Gad1/2*-positive, respectively, suggesting that the principal neurons of the vTT are glutamatergic.

It has previously been reported that the vTT has reciprocal connections with the olfactory bulb (OB), anterior piriform cortex (APC), and posterior piriform cortex (PPC) (*Igarashi et al., 2012*; *Luskin and Price, 1983a*; *Luskin and Price, 1983b*). In addition, the deep layers of the vTT receive top-down inputs from the medial prefrontal cortex (mPFC) (*Hoover and Vertes, 2011*). To further examine cortical areas projecting to the vTT, we injected a retrograde tracer, cholera toxin B subunit (CTB) conjugated with Alexa 555, into the mouse vTT (*Figure 6C*). A number of retrogradely labeled (CTB-positive) cell bodies were identified in the OB, APC, PPC, and mPFC. In contrast, CTB-positive cell bodies were rarely observed in the anterior olfactory nucleus (AON), which is located just dorsal to the vTT (*Figure 6D*).

To examine cortical regions that received axonal projections from vTT cells, we injected CTB into the mPFC, OB, AON, olfactory tubercle (OT), APC, and PPC. We then quantified retrogradely labeled CTB-positive cells in the vTT (*Figure 6E,F*). Many retrogradely labeled vTT cells were observed in mice that received CTB injections in the OB, AON, and APC. In contrast, only a small

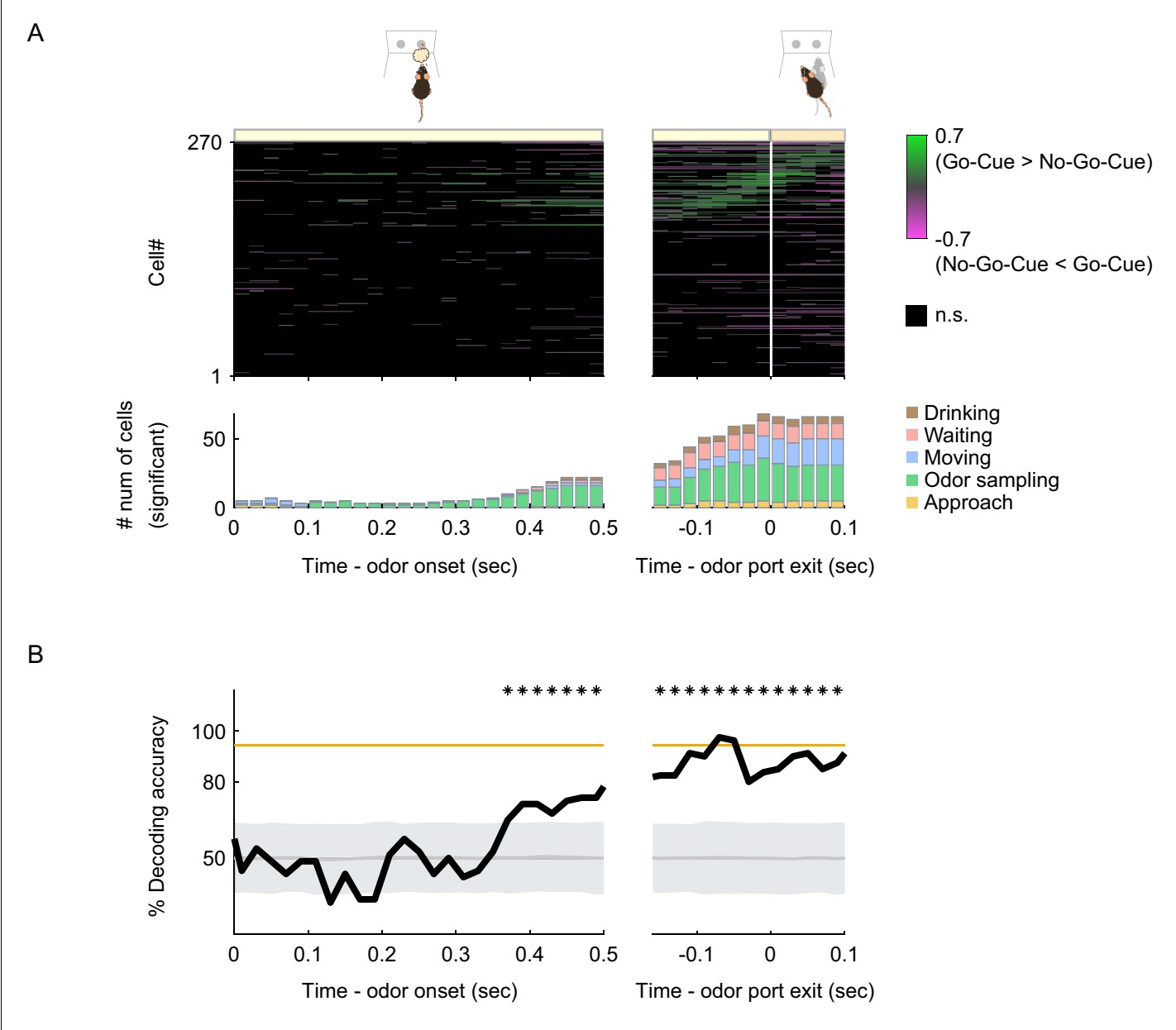

**Figure 5.** Behavioral context-specific activity of vTT cells. (**A**) Upper panel: auROC values for all cells around the odor sampling epoch in the odor-guided go/no-go task. Each row corresponds to one cell. auROC values (left, aligned to the odor valve opening; right, aligned to the odor port exit) were calculated for correct go trials versus correct no-go trials in sliding bins (width, 100 ms; step, 20 ms). These values were sorted based on the time of peak auROC values in the right graph. The vertical white line indicates the time of odor port exit. The color scale indicates significant auROC values (p<0.01, permutation test). The black boxes indicate bins with non-significant auROC values. Lower panel: the number of cells that exhibited significant auROC values (p<0.01, permutation test) for each time bin (orange, cell group tuned to the approach epoch; green, odor sampling epoch; blue, moving epoch; red, waiting epoch; brown, drinking epoch). (**B**) The time course of odor decoding accuracy in the odor-guided go/no-go task. A vector consisting of instantaneous spike counts for 270 neurons in a sliding window (width, 100 ms; step, 20 ms) was used as input for the classifier. Training of the classifier and testing were performed at every time point. The orange line indicates the level of behavioral performance. The gray line and shaded area indicate the mean ± 2 SD of control decoding accuracies calculated from 1000 trial-label-shuffled datasets. The shuffled datasets are provided in the *Figure 5—source data 1*. Top asterisks indicate accuracies greater than the mean + 2SD.

The online version of this article includes the following source data for figure 5:

**Source data 1.** Source data of the time course of odor decoding accuracy.

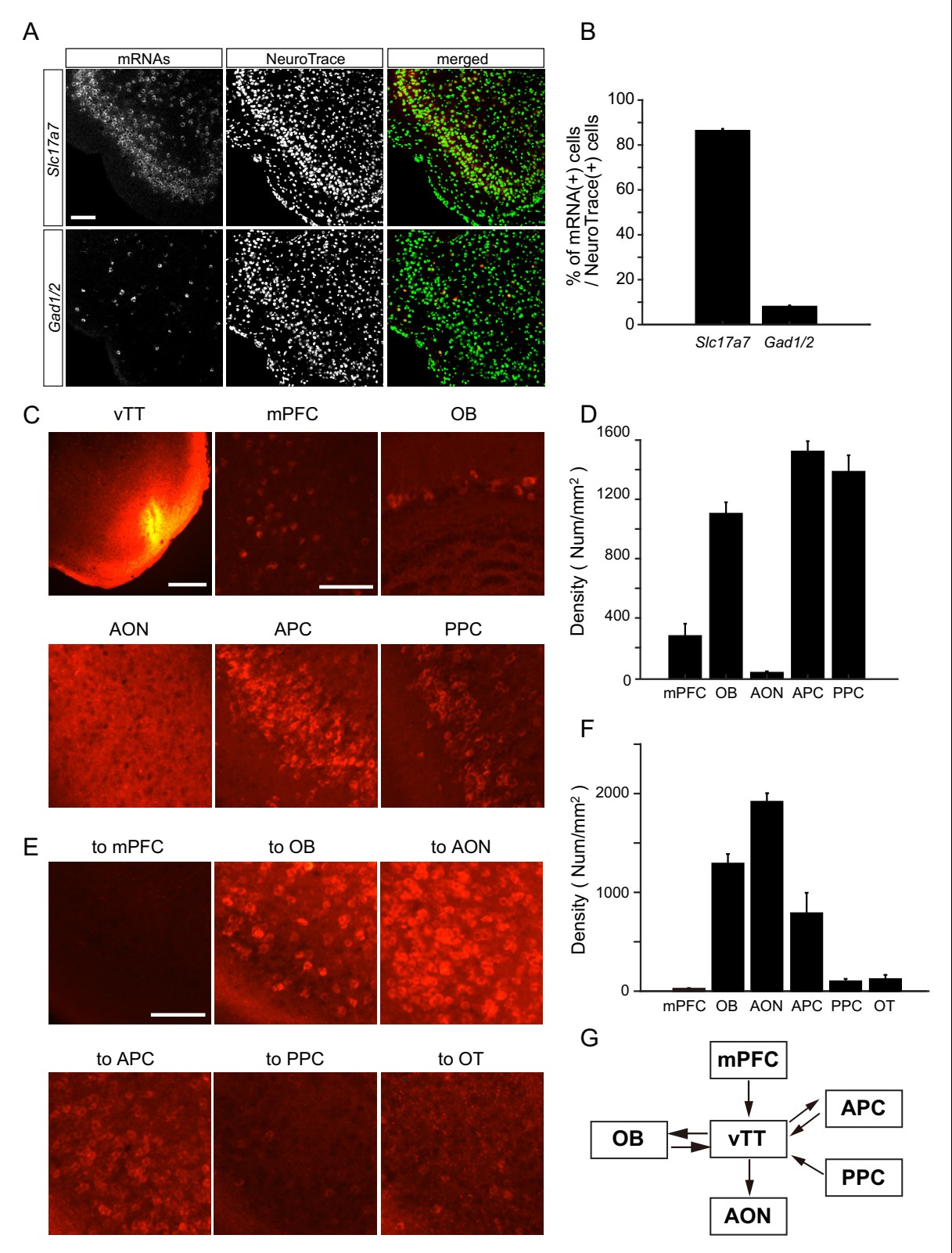

**Figure 6.** Cell types and connectivity patterns of vTT cells. (**A**) In situ hybridization of *Slc17a7* (upper panels) and *Gad1/2* (lower panels) mRNAs with Neurotrace staining of vTT cells. Scale bar, 100 μm. (**B**) Average percentages of *Slc17a7*-positive cells (left column) and *Gad1/2*-positive cells (right column) among Neurotrace-positive cells in the vTT (n = 3 mice). Error bars indicate SEM. (**C**) Upper left: Coronal section of the vTT after injection of Alexa 555-conjugated cholera toxin subunit B (CTB, red). Scale bar, 500 μm. The five other panels show CTB-labeled cells after CTB injections in the
*Figure 6 continued on next page*

Figure 6 continued

vTT. mPFC, medial prefrontal cortex; OB, olfactory bulb; AON, anterior olfactory nucleus; APC, anterior piriform cortex; PPC, posterior piriform cortex. Scale bar, 100 μm. (**D**) Average density of CTB-labeled cell bodies in each area (mPFC, n = 5 from three mice; OB, n = 5 from three mice; AON, n = 3 from two mice; APC, n = 5 from three mice; PPC, n = 5 from three mice). Error bars indicate SEM. (**E**) CTB-labeled vTT cells after injection of CTB into the mPFC (upper left), OB (upper middle), AON (upper right), olfactory tubercle (OT, lower left), APC (lower middle), and PPC (lower right). Scale bar, 100 μm. (**F**) Average density of retrogradely labeled CTB-positive cells in the vTT (n = 3 from two mice). Error bars indicate SEM. (**G**) Schematic diagram of vTT connectivity patterns. Arrows indicate axonal projections.

The online version of this article includes the following source data for figure 6:

**Source data 1.** Source data of the percentages of *Slc17a7*-positive cells and *Gad1/2*-positive cells in each animal.
**Source data 2.** Source data of the density of CTB-labeled cells in each area (D) and in the vTT (F).

---

number of retrogradely labeled vTT cells were observed in mice that received CTB injections in the OT and PPC. Furthermore, retrogradely labeled cells were scarce in the vTT following injections of CTB into the mPFC. These results suggest that, in addition to dense reciprocal connections with the OB, the vTT projects to the AON and APC, and receives top-down projections from the APC, PPC, and mPFC (*Figure 6G*).

## Discussion

In the present study, we reported recordings of neuronal activity in the vTT in freely behaving mice during the performance of odor-guided behaviors. We characterized the neural activities of the vTT cells in mice performing two types of odor-guided goal-directed behavioral tasks. Our results demonstrated characteristic tuning of individual vTT cells to specific temporal windows in the behavioral context of goal-directed tasks.

### Odor representation in vTT cells

In the odor-guided go/no-go task, 25% of vTT cells exhibited increased firing during the learned odor presentation phases (*Figure 1*). Although many of these vTT cells exhibited peak firing ~100 ms after odor onset, they did not encode learned odor differences during the odor presentation phases (*Figure 1E*). Firing rate differences in a subset of cells emerged just before the odor port exit during which mice performed behavioral choices (*Figure 5*). These features resembled the response pattern for learned odor-guided tasks observed in the piriform cortex (*Gadziola et al., 2020*; *Gire et al., 2013*). However, many vTT cells increased their firing rate before the odor presentation (*Figures 1D*, *2B* and *4B*), and individual vTT cells exhibited a particular tuning pattern characterized by peak tuning and tuning duration. Therefore, we hypothesized that the firing activity of vTT cells mainly reflected animal behavior and was dependent on task context.

### Context-dependent neural activity in the vTT

During the odor-guided go/no-go task used here (*Figure 2*), vTT cells exhibited a specific firing rate tuning to each behavioral epoch (e.g. moving to the odor port, odor-sampling in the odor port, moving to the water reward port, waiting for the reward, and obtaining the water reward) of the go trials. Furthermore, these cells tended to suppress their firing activity during epochs other than the most preferred epoch.

To evaluate context-dependent tuning of vTT cells, we employed a different odor-guided task. We identified three unique cellular subtypes based on firing pattern differences that occurred during the odor-guided eating/no-eating task (*Figure 3*). One cell type exhibited significant increases in firing during dish approach but tended to be silent during eating and food removal. A different subset of cells tended to fire more when the mouse approached the dish, during eating, and when the dish was removed. vTT cells also tended to suppress their firing during behavioral epochs other than their most preferred epoch.

Across tasks, we observed that tuning durations for each behavioral epoch correlated with the duration of behavioral epochs. In addition, the position of tuning peak times varied across the time course of the behavioral tasks (*Figure 4*). Further, many cells exhibited a short duration of activity at transitions between epochs, suggestive of a role in linking the two epochs (*Figure 4B*). Moreover,

we observed that vTT cells tuned to the odor-sampling epoch exhibited different activity between go and no-go trials just before odor port withdrawal (*Figure 5*), implying that individual vTT cells were tuned to a specific time window in task-modulated behavioral contexts rather than to a specific behavior.

Collectively, these results indicate that each vTT cell has a unique behavior- and context-dependent preference, demonstrated by tuning peak time and tuning duration. These properties of vTT cells may contribute to the representation of a series of specific behavioral context information (i.e. an episode) during odor-guided behaviors in the vTT. As vTT cells send their axons to other olfactory cortex areas such as the AON and piriform cortex (*Figure 6*; *Luskin and Price, 1983a*; *Luskin and Price, 1983b*), we speculate that the vTT may provide information to widespread olfactory areas regarding moment-to-moment changes in behaviors within certain behavioral contexts. Indeed, context-dependent modulation of neural activity is observed in mitral cells in the OB, an area that receives direct inputs from the vTT during the performance of odor-reinforcer association learning (*Kay and Laurent, 1999*).

We conjecture that behavioral context-dependent inputs from other higher-brain areas may contribute to the context-dependent activity of vTT cells because it is difficult to explain how behavioral context-dependent activity may otherwise be induced by bottom-up olfactory sensory input alone. The vTT receives direct top-down inputs from the mPFC and indirect inputs from other olfactory cortical areas (*Figure 6*; *Luskin and Price, 1983b*). In particular, mPFC neurons are implicated in contextual encoding when animals move between different environmental contexts (*Hyman et al., 2012*) and preferentially fire at a specific position in the trajectory of a maze during a working memory task involving odor place-matching (*Fujisawa et al., 2008*). Along with the evidence presented here, these findings suggest that context-specific activity of the mPFC may contribute to the generation of context-dependent activity in the vTT. In addition, Allen et al. recently reported that odor cue-induced reward-predicting responses were not restricted to the olfactory cortex but occurred throughout the brain, including sensory, motor, and prefrontal cortices, as well as subcortical regions (*Allen et al., 2019*). This suggests that context-dependent cells are located throughout the brain, and their activity may drive specific information processing modes in the brain. In this regard, the vTT may play an important role in entraining specific information processing modes in olfactory areas by acting as a hub, amplifier, or rectifier that sends context-dependent signals from the mPFC to broad olfactory areas.

Based on the firing patterns in the task behavioral epochs, we classified vTT cells into five and three cell types for the odor-guided go/no-go task and odor-guided eating/no-eating task, respectively. The tuning durations of these cells were strongly correlated with behavioral epoch durations (*Figure 4A*). However, the tuning durations of cells classified in the eating epoch showed a bimodal distribution, whereby some cells exhibited tuning durations that lasted for almost the entire eating epoch, and other cells exhibited short duration phasic responses. In addition, the neural activity of vTT cells was not always confined to a particular epoch. For example, many cells that increased their firing rates during the eating epoch started to fire prior to this epoch (*Figures 3* and *4B*). Moreover, individual vTT cells exhibited a unique tuning duration and tuning peak time (*Figure 4B*). We speculate that a subset of vTT cells are tuned to smaller scale behaviors or task elements involving moment-to-moment changes in a particular behavioral context. For instance, vTT cells, which exhibited peak firing at the transition between epochs, especially in the eating/no-eating task, may encode the initiation of movement.

## Putative encoding of internal states by vTT cells

During odor-guided goal-directed tasks, many vTT cells increased their firing rates prior to the onset of the most preferred behavioral epochs (*Figures 2*, *3*, *4* and *5*). These observations corroborate the idea that vTT cell activity is modulated both by signals during ongoing behavior as well as predictions about future behavior. Although a population of vTT cells increased their discharge rate during odor presentation in the odor-guided go/no-go task, their mean firing rate was not significantly different between go and no-go trials. Indeed, firing rate differences emerged just before mice exited the odor port. We speculate that vTT cell activity during the prediction of future behaviors was not driven directly by olfactory sensory inputs during ongoing behavior; rather, it was induced, directly or indirectly, by the piriform cortex and top-down inputs from higher cognitive and motivational centers in the prefrontal cortex (including mPFC) and amygdala. These regions play key roles in odor

identification, odor discrimination, motivation, reward prediction, and decision-making. vTT cells fired with different intensities according to distinct cue types during the eating epoch of the odor-guided eating/no-eating task (*Figure 3—figure supplement 4*). These response profiles may be underpinned by changes in the palatability of the reward flavors presented, since mice were exposed to both olfactory and taste stimuli during eating. Based on these results, we speculate that the firing patterns of vTT cells are influenced by both external contexts as well as internal states.

### Study limitations

Despite its strengths, the present study did not address whether or how top-down inputs influence vTT cells in particular behavioral contexts. Further, we did not elucidate how vTT cells integrate bottom-up and top-down inputs. Further work involving selective inhibition of top-down inputs or olfactory sensory inputs is needed to dissect the functional roles of vTT cells in odor-guided consummatory behaviors.

## Materials and methods

### Key resources table

| Reagent type (species) or resource | Designation | Source or reference | Identifiers | Additional information |
|---|---|---|---|---|
| Gene *Mus musculus* | Slc17a7 (*vglut1*) | NCBI Reference Sequence | NM_182993.2 | |
| Gene *Mus musculus* | Gad2 (*Gad65*) | NCBI Gene | 14417 | |
| Gene *Mus musculus* | Gad1 (*Gad67*) | NCBI Gene | 14415 | |
| Strain, strain background (*Mus musculus*) | Male C57BL/6 | Shimizu Laboratory Supplies Co., LTD and SLC Japan | RRID:MGI:5658686 | Wild-type mouse |
| Commercial assay, kit | DIG RNA Labeling Kit (SP6/T7) | Roche | Cat# 11175025910 | |
| Antibody | Anti-Digoxigenin-AP, Fab fragments (Sheep polyclonal) | Roche | Cat# 11093274910 RRID:AB_2734716 | (1:1,000) |
| Peptide, recombinant protein | CTB (Alexa FluorTM 555 Conjugate) | Thermo Fisher Scientific | Cat# C34776 | |
| Chemical compound, drug | HNPP fluorescence detection set | Roche | Cat# 11758888001 | |
| Chemical compound, drug | Eugenol | TOKYO CHEMICAL INDUSTRY Co., LTD. | Cat# A0021; CAS: 628-63-7 | |
| Chemical compound, drug | Vanilla essence | NARIZUKA Corporation | Cat# 100869 | |
| Chemical compound, drug | Almond essence | NARIZUKA Corporation | Cat# 100835 | |
| Chemical compound, drug | Amyl acetate | TOKYO CHEMICAL INDUSTRY Co., LTD. | Cat# A0021; CAS: 628-63-7 | |
| Software, algorithm | MATLAB_R2019a | The Mathworks, Inc | RRID:SCR_001622 | |
| Software, algorithm | KlustaKwik | PMID:25149694 | RRID:SCR_01448 | |

*Continued on next page*

*Continued*

| Reagent type (species) or resource | Designation | Source or reference | Identifiers | Additional information |
|---|---|---|---|---|
| Software, algorithm | Mclust-3.5 | AD Redish | http://redishlab. neuroscience. umn.edu/MClust/ MClust.html | |
| Other | Bpod r0.5 | Sanworks | RRID:SCR_015943 | Open-source control device designed for behavioral tasks. |
| Other | NeuroTrace 500/525 Green Fluorescent Nissl Stain | Thermo Fisher Scientific | Cat# N21480 | |
| Other | DiI | Molecular Probes | Cat# D3911 | 1 mM |
| Other | PermaFluor | Thermo Fisher Scientific | Cat# TA-030-FM | |
| Other | Polyimide-coated tungsten wire | California Fine Wire | | diameter 12.5 μm |

## Animal subjects

All experiments were performed on male C57BL/6 mice (9 weeks old; weighing 20–25 g) purchased from Shimizu Laboratory Supplies Co., Ltd. (Kyoto, Japan). Mice were individually housed in metal cages (13.5 × 23 × 17 cm) in a temperature-controlled environment with a 13 hr light/11 hr dark cycle (lights on at 8:00 and off at 21:00). Food and water were available ad libitum except during behavioral tasks. All experiments were performed in accordance with the guidelines for animal experiments at Doshisha University and with the approval of the Doshisha University Animal Research Committee.

## Odor-guided go/no-go task

For the odor-guided go/no-go task (*Figure 1A*), we used a behavioral apparatus controlled by the Bpod State Machine r0.5 (Sanworks LLC, NY), an open-source control device designed for behavioral tasks. The apparatus comprised a custom-designed mouse behavior box (14 × 17 × 15 cm) with two nose-poke ports on the front wall. The box was enclosed within a soundproof box (BrainScience Idea. Co., Ltd., Osaka, Japan) equipped with a ventilator fan to provide adequate air circulation and low-level background noise. Each nose-poke port was equipped with a white light-emitting diode (LED) and infrared photodiodes. Interruption of the infrared beam generated a transistor-transistor-logic (TTL) pulse, thus signaling the entry of the mouse head into the port. The odor delivery port was equipped with stainless steel tubing connected to a custom-made olfactometer (*Uchida and Mainen, 2003*). Eugenol and amyl acetate (Tokyo Chemical Industry Co., Ltd., Tokyo, Japan) were used as the go and no-go odor cues, respectively. These odors were diluted to 10% in mineral oil and further diluted 1:9 with airflow (1.8 L/min). Water reward delivery was based on gravitational flow, controlled by a solenoid valve (The Lee Company, CT), and connected via Tygon tubing to stainless steel tubing. The reward volume (6 μL) was determined by the duration of opening of the solenoid valve and was regularly calibrated.

For the odor-guided go/no-go task, mice (n = 6) were placed on a water restriction schedule of 1–2 mL/day with daily body weight monitoring to ensure that their body weight remained within 80% of their body weight prior to restriction. During the training sessions, mice learned to obtain water rewards at the left water port, move from the right odor port to left odor port, and associate odor cues with the correct action. Each trial began with illumination of the LED light at the right odor port, which acted as a signal for the mouse to nose poke into that port. A nose poke into the odor port resulted in the delivery of one of the two odor cues for 500 ms. Mice were required to maintain their nose pokes during odor stimulation to sniff the odor. After odor stimulation, the LED light was turned off and the mice could withdraw their nose from the odor port. If eugenol odor (go

odor cue) was presented, the mice were required to move to and nose poke into the left water reward port within a timeout period of 2 s. At the water port, the mice were required to maintain their nose poke for 300 ms before water delivery began. Next, 6 μL of water was delivered as a reward. If amyl acetate odor (no-go odor cue) was presented, the mice were required to avoid entering the water port for 2 s following odor stimulation. The average inter-trial interval (ITI) was 3 s.

## Odor-guided eating/no-eating task

For the odor-guided eating/no-eating task (*Figure 3A*), mice (n = 6) were placed on a food restriction schedule of 3–4 g/day with daily body weight monitoring to ensure that their body weight remained within 80% of their body weight prior to restriction. During the training sessions, mice learned to associate odors with a sucrose reward. Training was conducted in a plastic cage (38.5 × 33.5 × 18 cm, CLEA Japan Inc, Tokyo, Japan) which contained virgin pulp bedding (SLC, Inc, Shizuoka, Japan). The cage was monitored with a recording camera in a soundproof room with a ventilator fan that provided air circulation and a low-level of background noise. Mice were presented with 1–2 g of granulated sugar on a Petri dish with holes which contained a filter paper (2 × 2 cm) soaked with odor (40 μL). The dish was covered with bedding at an arbitrary position in the cage. The odor cues were eugenol (Tokyo Chemical Industry Co., Ltd., Tokyo, Japan) and vanilla essence (NARIZUKA Corporation, Tokyo, Japan), diluted to 10% in mineral oil. During initial training sessions, odor trials were randomly presented. After learning the association between the dish, odor, and sugar reward, mice would approach and touch the dish, dig through the overlying bedding, and proceed to consume the sucrose. As we used large beddings (the size of each bedding about 10 × 5 mm), the bedding did not mix completely with the sucrose. They could get mixed up somewhat, but the sucrose was covered with a small amount of new bedding before each trial so that the sucrose was not able to be accessed without digging the bedding. Mice required at least seven initial training sessions.

After the initial training sessions, mice were trained to associate an almond essence odor (NARIZUKA Corporation), diluted to 10% in mineral oil with an aversive consequence (malaise) (*Raineki et al., 2009*). This was performed by allowing mice to approach the dish with the almond essence odor and consume the sucrose in the dish for 1 hr, after which mice were intraperitoneally injected with 0.5 M of lithium chloride (LiCl, 0.01 mL/g). After receiving the LiCl injection, mice would continue to approach the dish with the almond odor but did not consume the sucrose (no-eating response). No-eating responses were established in a single session.

After conditioned aversion training, performance in the odor-guided eating/no-eating task was examined. Sucrose was randomly presented on a dish with one of three different odor cues (eugenol, vanilla essence, or almond essence). The almond essence odor was presented with a 20% probability. Powder chow was also presented on a dish with a 20% probability among trials. Approximately 6 s after the mice began to eat the dish was abruptly removed. The inter-trial interval (ITI) was explicitly not determined to increase the unpredictability of events. Correct trials were defined as approach and consumption for eugenol, vanilla essence, and powder chow trials. For almond essence trials, correct trials were defined as approach but no consumption or no approach for at least 30 s after the dish was presented. After the mice were well trained, their behavioral accuracy remained above 80% throughout the session. Mice were subjected to 40–60 daily trials of the task.

## Electrophysiology

Mice were anesthetized with medetomidine (0.75 mg/kg i.p.), midazolam (4.0 mg/kg i.p.), and butorphanol (5.0 mg/kg i.p.), and implanted with a custom-built microdrive of three or four tetrodes in the vTT (2.6 mm anterior to bregma, 0.4 mm lateral to the midline, 4.0 mm from the brain surface). Individual tetrodes consisted of four twisted polyimide-coated tungsten wires (California Fine Wire, single wire diameter 12.5 μm, gold-plated to less than 500 kΩ). The tips and sides of the tetrodes were coated with DiI (Molecular Probes, OR). Two additional screws were threaded into the bone above the cerebellum for reference. The electrodes were connected to an electrode interface board (EIB-18, Neuralynx, MT) on a microdrive. A microdrive array was fixed to the skull with LOCTITE 454 (Henkel Corporation, Düsseldorf, Germany). After the completion of surgery, mice received atipamezole (0.75 mg/kg i.p.) to reverse the effects of medetomidine and enable a shorter recovery period.

Analgesics were administered post-surgery (ketoprofen, 5 mg/kg, i.p.). Behavioral training resumed at least 1 week after surgery.

Electrical signals were obtained with either a Cheetah recording system (Neuralynx) or open-source hardware (Open Ephys). For unit recordings, signals were sampled at 32 kHz in NeuraLynx or 30 kHz in Open Ephys and band-pass filtered at 600–6,000 Hz. After each recording, tetrodes were lowered by 20, 40, or 75 μm more by turning a screw of the microdrive to obtain new units.

## Data analyses

All data analyses were performed using built-in software in MATLAB 2019a (The Mathworks, Inc, MA).

### Behavioral data analyses

For the odor-guided go/no-go task, the accuracy rate was calculated as the total percentage of successful responses in the go and no-go trials in a session. Mice performed up to 600 trials in each session per day. For the odor-guided eating/no-eating task, the accuracy rate was calculated as the average percentage of trials in which mice successfully consumed sucrose paired with eugenol or vanilla essence odor, successfully consumed the powder chow, or did not consume sucrose paired with almond essence odor. Mice performed 40–60 trials in each session per day.

### Spike sorting

Spikes were sorted into clusters offline based on their waveform energy, peak amplitudes, and first principal components from the four tetrode channels using an automated spike-separation algorithm KlustaKwik (K.D. Harris). Resulting classifications were corrected and refined manually with MClust software (A.D. Redish). Clusters were considered single units only when the following criteria were met: (1) refractory period (2 ms) violations were less than 0.2% of all spikes and (2) isolation distance, estimated as the distance from the center of the identified cluster to the nearest cluster based on the Mahalanobis distance, was more than 20. To examine the behavioral correlates of vTT cell firing patterns, we selected vTT cells with average firing rates during the trials that were greater than 0.3 Hz for further analyses.

### Spike train analyses

In the odor-guided go/no-go task, neural and behavioral data were synchronized by inputting each event timestamp from the Bpod behavioral control system into the electric signal recording system. To calculate firing rates during tasks, peri-event time histograms (PETHs) were calculated using a 20 ms bin width and smoothed by convolving spike trains with a 60 ms wide Gaussian filter (bottom lines in *Figure 1C* and *Figure 2—figure supplement 1A*). In the odor-guided eating/no-eating task, timestamps were acquired for each event (onset of dish approach, initial dish contact, and dish removal) from frames of recorded behavioral videos, which were synchronized to spike data. To calculate cell firing rates during tasks, PETHs were calculated using a 500 ms bin width and smoothed by convolving spike trains with a 500 ms wide Gaussian filter (*Figure 3—figure supplement 2A*).

### Event-aligned spike histograms

To examine the relationship between firing rate changes among individual vTT cells and the development of behavioral epochs in behavioral tasks, EASHs were generated (*Ito and Doya, 2015*). As behavioral epoch durations varied for each trial, median epoch durations were first calculated. In the odor-guided go/no-go task, the median duration of odor sampling epochs (from the onset of the go odor cue to exiting the odor port) was 839 ms, and the median duration of moving epochs (from odor port exit to water port entry) was 455 ms. Spike timing during each epoch and for each trial was linearly transformed to correspond with the median duration of each behavioral epoch. The number of spikes in each epoch was preserved. We defined the approach epoch (500 ms before odor port entry), waiting epoch (300 ms reward delay, from entry into the water port to the onset of water reward), and drinking epoch (1000 ms after the onset of the water reward). These epochs were not applied to the transformation because their durations did not change across trials. In this manner, a regular raster plot was transformed into event-aligned raster plots (left plots in *Figure 2A* and *Figure 2—figure supplement 1B*). An EASH was subsequently calculated using a 20 ms bin

width and smoothed by convolving spike trains with a 60 ms wide Gaussian filter from the raster plot.

In the odor-guided eating/no-eating task, the median duration of the approach epoch (from the start of approach behavior to contacting the dish) was 2.57 s, and the median duration of the eating epoch (from the start of consumption to food removal) was 5.44 s. Spike timing during each epoch of each trial was linearly transformed into a corresponding median duration for each epoch. The number of spikes in each epoch was preserved. We defined the food removal epoch (2 s after the start of food removal), which was not applied to this transformation because the duration of this epoch did not change across trials. Thus, regular raster plots were transformed into event-aligned raster plots (left plots in *Figure 3B* and *Figure 3—figure supplement 2B*). An EASH was subsequently calculated using a 500 ms bin width and smoothed by convolving spike trains with a 500 ms wide Gaussian filter from the raster plots.

## ROC analyses

To quantify firing rate changes, we used an algorithm based on ROC analyses that calculates the ability of an ideal observer to classify whether a given spike rate was recorded in one of two conditions (e.g. during go or no-go odor cue presentation) (*Felsen and Mainen, 2008*). We defined an auROC equal to 2 (ROC area – 0.5), with the measure ranging from –1 to 1, where –one signified the strongest possible value for one alternative and one signified the strongest possible value for the other.

The statistical significance of ROC analyses was determined using a permutation test. For this test, ROC curves were recalculated after all firing rates were randomly assigned to either of the two groups (e.g. go-cue trial group and no-go-cue trial group) arbitrarily. This procedure was repeated a large number of times (500 times for all analyses) to obtain a distribution of values. The fraction of random values exceeding the actual value was then calculated. Based on this quantification, the auROC value for each cell was calculated along with a time course for the go/no-go or eating/no-eating tasks in sliding windows (in overlapping 100 ms windows starting every 20 ms for the go/no-go task and overlapping 500 ms windows starting every 100 ms for the odor-guided eating/no-eating task). The auROC values during the go or no-go trials were then calculated by quantifying firing rate changes from baseline (*Figures 1D* and *2A*). A similar procedure was employed for the eating trials by quantifying firing rate changes from baseline (*Figure 3B*) and for go trials by quantifying firing rate changes relative to no-go trials (*Figure 5A*). For all analyses, significance was tested with α = 0.01. Only cells with a minimum number of 10 trials for each analyzed condition were included in these analyses.

## Evaluation of neural tuning to specific behaviors

To determine the timepoint at which each vTT cell increased its firing rate the most and the duration of increased firing during the task, two measures were calculated from each auROC value (*Figure 2—figure supplement 3*):

1. Tuning peak time: the time corresponding to the peak of the significant points when a cell significantly increased its firing rate from baseline for five or more consecutive bins (p<0.01, permutation test)
2. Tuning duration: the duration, including the tuning peak time, for which increased firing was significant (p<0.01, permutation test)

In the odor-guided go/no-go task, we classified vTT cells with tuning peak times (n = 217 cells) into five groups based on the epoch in which tuning peak time occurred during correct go trials. Of these, 7% of all cells had tuning peak times in the approach epoch, 24% in the odor-sampling epoch, 23% in the moving epoch, 11% in the waiting epoch, and 15% in the drinking epoch.

In the odor-guided eating/no-eating task, we classified vTT cells with tuning peak times (n = 192 cells) into three groups based on the epoch in which tuning peak time occurred during the eating trials. In total, 19% of all cells had tuning peak times during the approach epoch, 20% during the eating epoch, and 12% during the food removal epoch.

### Relative distributions of significant responses

The distribution of behavioral epoch-specific firing for each cell type (yellow and blue shadows in *Figures 2B* and *3C*) was calculated as follows (*Figure 2—figure supplement 4*). First, for each cell group, the number of cells with significant excitatory (or suppressed) firing during each time bin for each task was calculated. Their distribution was estimated using Gaussian kernel smoothing via the fitdist function in MATLAB with a bandwidth of 100 ms in the odor-guided go/no-go task and 500 ms in the odor-guided eating/no-eating task. Second, the distribution of the significant excitatory (or suppressed) firing was compared with the cell-shuffled data for each cell group. To calculate the 99% confidence intervals of this distribution, we ran 1000 iterations in which cellular group identities were randomized. We then calculated the distribution of significant excitatory (or suppressed) firing of each cell group, as above. For each bin in the tasks, the edges of the confidence interval were at the 0.5$^{th}$ and 99.5$^{th}$ percentiles of the distribution, as calculated from the cell-shuffled data (gray-shaded area in *Figure 2—figure supplement 4C*). The significance of activity patterns for each cell group was determined by comparing the standard deviations of the data with those obtained from the cell-shuffled data, in which shuffling was performed 1000 times by randomizing the cell selection.

### SVM decoding analyses

A support vector machine (SVM) algorithm with a linear kernel as a classifier (*Cury and Uchida, 2010*; *Miura et al., 2012*) and a Matlab function (fitcsvm) were used for analyses. All analyses were conducted on trial data pooled across animals. A matrix containing concatenated firing rates for each trial and each cell was the input for the classifier. The matrix dimensions were the number of cells by the number of trials. To avoid over-fitting, k-fold cross-validation (k = 10) was used to calculate the decoding accuracy of trial type discriminations. To compute decoding accuracy, 40 trials for each trial type (from the start of the session) were selected as the data set. Next, the data set was partitioned into 10 equal parts; one part was used for testing, and the remaining parts were used for training the classifier. This process was repeated 10 times to test each individual part. The mean value of the accuracy was then used for decoding accuracy. To compute the decoding accuracy of a 100 ms bin window (step: 20 ms), the classifier was trained and tested with a 100 ms bin window (step: 20 ms). In order to verify the statistical significance of the computed accuracy, we estimated the accuracy distribution on label-shuffled data using 1000 repetitions of a random selection from the data set.

### Statistical analysis

Data were analyzed in Matlab 2019a. Statistical methods for each analysis are described above, in the results section, or in the figure legends. The Tukey-Kramer method was applied for significance tests with multiple comparisons. Although sample sizes in this study were not pre-determined with sample size calculations, they were based on previous research in the olfactory cortex field (*Manabe et al., 2011*; *Miura et al., 2012*). Randomization and blinding were not employed. Biological and technical replicates for the goal-directed tasks are depicted in *Tables 1* and *3*. Biological replicates for the histological studies are described in the figure legends.

## Histology

After recording, mice were deeply anesthetized with intraperitoneal injection of sodium pentobarbital. Electric lesions were induced using 10–20 µA direct current stimulation for 5 s using one of the four tetrode leads. Mice were perfused transcardially with phosphate-buffered saline (PBS) followed by 4% paraformaldehyde (PFA). Brains were removed from the skull and post-fixed in PFA. Brains were sliced into 50-µm-thick coronal sections. The electrode tracks were first assessed by tracking the fluorescence of DiI. After staining with cresyl violet, electrode tracks and recording positions were determined in reference to the mouse brain atlas developed by *Paxinos, 2004*.

## In situ hybridization

Adult male mice (n = 3) were used. Digoxigenin-(DIG)-labeled RNA probes for *Slc17a7* and *Gad1/2* were generated using an in vitro transcription kit (Roche, Basel, Switzerland), according to the manufacturer's protocol using plasmids generously provided by Drs. Katsuhiko Ono and Yuchio Yanagawa

(*Asada et al., 1997*; *Makinae et al., 2000*; *Ono et al., 2008*). Brain sections (20-μm-thick) were mounted on glass slides (CREST, Matsunami, Osaka, Japan) using a paintbrush and dried overnight in a vacuum desiccator. The dried sections were then fixed in 4% PFA, digested with Proteinase K (10 μg/mL) for 30 min, and post-fixed in 4% PFA. After prehybridization, the sections were incubated overnight at 65°C with DIG-labeled RNA probes. After stringent washing, sections were blocked with 1% blocking reagent (11096176001, Roche) in TNT for 1 hr. Subsequently, sections were incubated overnight at 4°C with alkaline phosphatase-conjugated anti-DIG antibody (1:1,000; Roche). Sections were then washed three times in TNT and once in TS 8.0 (0.1 M Tris-HCl, pH 8.0, 0.1 M NaCl, 50 mM MgCl$_2$). Finally, alkaline phosphatase activity was detected using an HNPP fluorescence detection set (11758888001, Roche), according to the manufacturer's instructions. After three 30 min incubations, sections were washed with PBS. Finally, sections were counterstained with NeuroTrace green (Thermo Fisher Scientific, MA) and mounted in PermaFluor (Thermo Fisher Scientific).

## Retrograde tracing

Adult male mice were anesthetized with medetomidine (0.75 mg/kg i.p.), midazolam (4.0 mg/kg i. p.), and butorphanol (5.0 mg/kg i.p.), and then placed in a stereotaxic apparatus (SR-5M, Narishige, Tokyo, Japan). Injections were conducted with a syringe pump (UltraMicroPump III, WPI, FL) connected to a Hamilton syringe (RN-1701, Hamilton, Nevada) and a mounted glass micropipette with 50 μm tip diameter connected to the syringe with an adaptor (55750–01, Hamilton).

CTB conjugated with Alexa 555 (Thermo Fisher Scientific) was unilaterally or bilaterally injected (300 nL at 100 nL/min) into the mPFC (A/P, 2.4 mm; M/L, 0.4 mm from bregma; D/V, 1.0 mm from the brain surface; n = 3 from two mice), OB (A/P, 4.3 mm; M/L, 0.8 mm from bregma; D/V, 1.5 mm from the brain surface; n = 3 from two mice), AON (A/P, 2.8 mm; M/L, 1.3 mm from bregma; D/V, 2.6 mm from the brain surface; n = 3 from two mice), OT (A/P, 1.5 mm; M/L, 1.0 mm from bregma; D/V, 4.7 mm from the brain surface; n = 5 from three mice), APC (A/P, 2.3 mm; M/L, 1.8 mm from bregma; D/V, 3.4 mm from the brain surface; n = 4 from four mice), or PPC (A/P, −1.5 mm; M/L, 3.6 mm from bregma; D/V, 4.5 mm from the brain surface; n = 3 from two mice), or vTT (250 nL; A/P, 0.3 mm tilted 30°; M/L, 0.4 mm from bregma; D/V, 4.6 mm from the brain surface; n = 3 from two mice). After surgery, mice received atipamezole (0.75 mg/kg i.p.) and ketprofen (5 mg/kg, i.p.). One week later, mice were deeply anesthetized and perfused with saline followed by 4% PFA under anesthesia. Brains were sliced into 50-μm-thick coronal sections.

## Microscopy

Sections were examined with a confocal laser microscope (FV1200, Olympus, Tokyo, Japan) and a bright-field and fluorescent microscope (Zeiss, Oberkochen, Germany).

## Acknowledgements

We thank Kensaku Mori for valuable discussion and advice. And we thank Hideki Tanisumi for providing mouse illustrations in Figures. KS was supported by JSPS KAKENHI Grant Numbers 18J21358. YT was supported by JSPS KAKENHI Grant Numbers 19J20733. YS was supported by JSPS KAKENHI Grant Numbers 16H02061, 18H05088. HM was supported by the Takeda Science Foundation, Narishige Neuroscience Research Foundation, and JSPS KAKENHI Grant Numbers 25135708, 16K14557.

## Additional information

### Funding

| Funder | Grant reference number | Author |
| --- | --- | --- |
| Japan Society for the Promotion of Science | Grant-in-Aid for JSPS Fellows 18J21358 | Kazuki Shiotani |
| Japan Society for the Promotion of Science | Grant-in-Aid for Challenging Exploratory Research 16K14557 | Hiroyuki Manabe |

| | | |
|---|---|---|
| Japan Society for the Promotion of Science | Grant-in-Aid for Scientific Research on Innovative Areas 25135708 | Hiroyuki Manabe |
| Takeda Science Foundation | | Hiroyuki Manabe |
| Narishige Neuroscience Research Foundation | | Hiroyuki Manabe |
| Japan Society for the Promotion of Science | Grant-in-Aid for Scientific Research(A) 16H02061 | Yoshio Sakurai |
| Japan Society for the Promotion of Science | Grant-in-Aid for Scientific Research on Innovative Areas 18H05088 | Yoshio Sakurai |
| Japan Society for the Promotion of Science | Grant-in-Aid for JSPS Fellows 19J20733 | Yuta Tanisumi |

The funders had no role in study design, data collection and interpretation, or the decision to submit the work for publication.

### Author contributions
Kazuki Shiotani, Conceptualization, Data curation, Software, Formal analysis, Funding acquisition, Validation, Investigation, Visualization, Methodology, Writing - original draft, Project administration, Writing - review and editing; Yuta Tanisumi, Conceptualization, Data curation, Software, Formal analysis, Funding acquisition, Validation, Investigation, Visualization, Methodology, Writing - original draft, Writing - review and editing; Koshi Murata, Data curation, Formal analysis, Visualization, Writing - review and editing; Junya Hirokawa, Software, Supervision, Writing - review and editing; Yoshio Sakurai, Supervision, Funding acquisition, Writing - review and editing; Hiroyuki Manabe, Conceptualization, Data curation, Formal analysis, Supervision, Funding acquisition, Validation, Investigation, Visualization, Methodology, Writing - original draft, Project administration, Writing - review and editing

### Author ORCIDs
Kazuki Shiotani (iD) http://orcid.org/0000-0001-5596-5609
Junya Hirokawa (iD) http://orcid.org/0000-0003-1238-5713
Hiroyuki Manabe (iD) https://orcid.org/0000-0002-3910-4849

### Ethics
Animal experimentation: Animal experimentation: Animal experiments were approved and performed in accordance with the guidelines for the care and use of laboratory animals established by the Committee for Animal Care (Permit Number: A15089, A16013, A17007, A18011) of Doshisha University. All efforts were made to minimize animal suffering and the number of animals used.

### Decision letter and Author response
Decision letter https://doi.org/10.7554/eLife.57268.sa1
Author response https://doi.org/10.7554/eLife.57268.sa2

## Additional files

### Supplementary files
• Transparent reporting form

### Data availability
All data generated or analysed during this study are included in the manuscript and supporting files. Source data files have been provided for Figure 2, 3, 5 and 6.

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
