## [Decision Letter]

**Acceptance summary:**

The ventral tenua tecta (vTT), one of the olfactory cortical areas, remains one of the most poorly characterized areas in the brain. This study provides the first characterization of task-related activities in the vTT and provides interesting findings. The observation that vTT neurons are very poorly tuned to odors but change their activity in different task events, tiling the entire task period, is surprising and important for further characterization of this brain area.

**Decision letter after peer review:**

Thank you for submitting your article "Tuning olfactory cortex ventral tenia tecta neurons to a distinct scene in the context of goal-directed behavior" for consideration by *eLife*. Your article has been reviewed by Laura Colgin as the Senior Editor, a Reviewing Editor, and two reviewers. The following individuals involved in review of your submission have agreed to reveal their identity: Daniel W Wesson (Reviewer #3).

The reviewers have discussed the reviews with one another and the Reviewing Editor has drafted this decision to help you prepare a revised submission.

Summary:

The ventral tenia tecta (vTT) is one of the olfactory cortical areas, about which we know very little. In this study, Shiotani and colleagues characterized the activity of vTT neurons in mice performing two types of odor-guided behavioral tasks. The first task is a nose-poke based Go/No-Go task using odor cues. Although vTT is considered an olfactory area, receiving direct inputs from the olfactory bulb, very few neurons responded to odors in an odor-specific manner. Instead, many neurons were excited at a specific task epoch defined by salient task events (approaching, odor sampling, choice movement, waiting for reward, and drinking a water reward). Most of the neurons that responded during odor sampling did not distinguish Go versus No-go cues. Next, the authors developed a new task where different odors predicted a food reward (sucrose) followed by nothing or a food reward followed by a LiCl injection (aversive consequence). Similar to those findings in the first task, the authors observed that neuronal activations are associated with a specific task epoch although the two tasks involved behaviors on different timescales (a fraction of a second versus seconds). Based on these results, the authors suggest that vTT neurons can be regarded as "scene cells".

Essential revisions:

This is likely the first study that characterized the activity of vTT neurons in behaving animals. The reviewers appreciated the novelty of the study, and they found various observations interesting. However, they raised various issues that need to be addressed before publication of this study in *eLife*. First, the main conclusion that vTT neurons are "scene cells" warrants more careful evaluations. The concept of "scene cells" is neither well-defined nor sufficiently supported by the presented data. For instance, reviewer #1 points out that a "scene cell" suggests that the activity should be relatively stable during each task event, but many neurons appear to exhibit transient activations at transitions between different task events. Reviewer #2 suggests removing the term altogether. We would like to see more concrete definitions and justification of this term. Alternatively, the authors might want to seek for different terms that better summarize the observed activity or not "naming" these cells. Additionally, the reviewers raised a number of technical and interpretational issues as described below. Although addressing these issues likely do not require additional experiments, we would like the authors to address these issues in a revised manuscript.

Reviewer #1:

This manuscript describes neural representations of vTT neurons during odor-guided tasks. The authors use tetrodes to record from populations of vTT neurons while mice are performing either a go/no go two-odor discrimination task, or a task in which mice search for and then eat odorized chow. Little is known about what vTT neurons do, and the authors' results are novel and convincing, and will be interesting to the field.

Essential revisions:

1) The authors measured auROC values and described these frequently as "normalized firing rates" (e.g. Figure 1D, E etc). The auROC is a measure of the discriminability of two distributions and not of response strength. If the authors want to talk about the size and strength of the responses then they should recalculate their data using either raw spike rates, or z-scores if they want to normalize across cells. If they want to simply make the point that response distributions are different then auROC analyses are fine.

The temporal resolution of the analyses is pretty course. If the authors use smaller time bins for the physiology as well as the behavioral analyses they may uncover additional dynamics within each epoch. (For the go/no-go assays they use 20 ms bins and a 60 ms smoothing kernel; for the eating/no-eating assay they use 500 ms binning and 500 ms smoothing.) There are a number of reasons why this may be valuable:

2) First, the authors want to convince us that vTT encodes behavioral scenes, and Figure 4A indicates a strong correlation between the behavioral and neural activity durations. However, Figure 4B indicates that a very large fraction of the recorded cells actually fire at the transitions between epochs. If this is the case, then can we really say that cells in vTT encode behavioral scenes?

3) Second, as very little is known about vTT additional insight might be gained by using a more fine-grained analysis. For example, in Figure 4A, the authors show a strong correlation between median behavioral duration and median firing activity duration. However, the eating epoch, which drives the strong correlation in 4A, has a bimodal distribution of neural activity durations, with many cells having brief responses (< 1 second) and other cells having responses that last for most of the eating epoch. I suspect that there could be multiple classes of neurons (e.g. a transient food odor population and a sustained eating population) embedded within their relatively small number of existing classes.

4) In the Discussion section, the authors propose that vTT neurons may be coincidence detectors for sensory and top-down inputs. Their tracing studies, which are consistent with older studies, indeed indicate that vTT receives mPFC input, which could be the source of top-down input. However, there are at least two reasons why their results are inconsistent with this model. First, only 25% of their cells were odor responsive yet most of the other cells responded at one or another of the behavioral epochs, even when there was ostensibly no (strong) olfactory input; second, the odor responses to two different odors were essentially identical, Both of these are difficult to reconcile with a system that is strongly driven, or even contingent on direct (from bulb) olfactory input, and instead suggest a circuit that is predominantly driven by top-down input and has surprisingly little (if any) bottom-up sensory drive. For example, do the authors think that vTT neurons would respond similarly or differently to an auditory go/no-go task?

Reviewer #2:

This is a compelling and well-written manuscript on a population of neurons which has received to date the least attention of all systems in the 'olfactory cortex'. The ventral tenia tecta has an unknown function and is quite small thus making monitoring activity of these cells technically difficult. Here, the authors uncovered that vTT neurons are recruited in mice during engagement in two different odor-guided behavioral tasks, and that the modulation occurs by separate groups of neurons, during different behavioral epochs of the tasks. The authors have shined light on something novel in the context of an underexplored brain region (a unique opportunity which is exciting) and did so in two different tasks which helps show the generalization of the outcomes.

Essential revisions:

1) The authors include data on the chemical identity of vTT neurons (not those recorded from necessarily) and also their connectivity which is tangential to the main story and does not directly support or clarify any pressing matters needed to support major conclusions. We appreciate what the authors are trying to do with these data (helping readers know where input arrives from, and what the output may influence), but these data are not worthy of being included in this manuscript and do not add to the main message in a meaningful way. The main story is the unit data – which is clear, strong, and on its own makes an excellent paper.

2) There is pervasive speculation in this paper and also instances of hypotheses being foreshadowed as to be tested, but never acted upon.

Regarding the latter, the authors state the hypothesis in the Introduction that higher order top-down inputs are generating changes in the vTT – yet this hypothesis is never tested in this paper. This is never tested and the anatomy in question above (point 1) does not test that hypothesis neither.

We are all for some appropriate scholarly speculation in Discussion section, but in many cases it reads as if the authors are possibly writing about data they have yet to acquire or just did not include. The speculation in the Discussion section that learned odor information about certain contexts is handled "mainly" by the vTT is not warranted by these data which only investigated [albeit compellingly and carefully] vTT activity. At a simplistic level, it is not intuitive that such a small brain region would exert major behavioral influence in contrast to other regions like PCX or OT which also encode learned odor information and have vast connectivity with downstream structures important with affect and behavior. Also, there are many mentions of top-down inputs to vTT being important mediators of the task modulation shown by the authors. Yet this is never shown and are not necessary for the main story of the paper. These findings are interesting in themselves and worthy of publication without having to try to make some circuitry story embedded within. The authors need to seriously restrict speculation about top-down input and also again, remove hypotheses about top-down inputs being integral for the unit activity they report. Focusing the discussion on why the results at hand matter and how they advance the field would be appreciated. Including more discussion on how these results compare/contrast with behavioral modulation of PCX, OT, and OB units would be helpful to appreciate if and how vTT is unique.

3) The authors define the modulated vTT cells as "scene" cells: "We found that the firing of individual vTT neurons during odor-guided goal-directed behaviors was highly tuned to distinct behavioral "scenes" (i.e., distinct task-elements that occurred in relation to the flow of goal-directed tasks, with each task-element occurring in a specific behavioral context). This is an attractive term, sure, but it is arguable then that almost all single units monitored in the olfactory system, or many other brain regions for that matter, could also be called "scene" cells and thus packaging them with this name is not optimal. For instance, in 1980 Karpov recorded mitral and tufted cells from behaving rabbits as they approached boxes containing odor which they were motivated for. The authors reported spiking upon approach, upon odor, and upon food intake. These results [while certainly not with the rigor in analyses as the authors used in the present manuscript] indicate that mitral cells encode "distinct task-elements that occurred in relation to the flow of goal-directed tasks, with each task-element occurring in a specific behavioral context" – also fitting the authors definition of a "scene cell". We could say the same if we compare and contrast the work herein to many other awake unit recording papers whether in the context of odors or not, where the authors report, of all task modulated units, some are modulated upon anticipation, instrumental responding, stimulus delivery, reward seeking, reward acquisition, etc. There is a possibly latent reason in every one of these papers that authors do not try to name the units with a descriptor – ultimately the descriptor would become useless in the field. This is unlike descriptors bestowed upon other cell types, like "head direction" cells – in these cases one must do many manipulations while recording from the same cell to prove head-direction encoding. Fortunately, this is an easy fix for the authors to remove "scene" throughout the paper. Including a succinct discussion on this concept however is very welcome.

4) It seems the use of the decoding is not fully taken advantage of by the authors. The decoding should be compared to a shuffled baseline, or statistically validated somehow, to see at what point the classification accuracy is significantly greater than chance… in this paper the authors use it only to say that when trained on all units, the classifier increases in accuracy before odor port exit (so it tracks with the behavior and not the odor itself). But wouldn't it be more compelling to see how classification accuracy differs across the populations they identified? This would strengthen the conclusions that these different populations of modulated neurons are handling information differently. This we will leave at the will of the authors though since, while likely informative, it may delay publication.

5) Details on adjustment of tetrode locations between recording sessions is needed to ensure redundant sampling of units did not occur. The authors state some mice were recorded from for 18 sessions (in table), but if the tetrodes were driven enough to rule out resampling the same units on subsequent days, but how 18 movements can occur in the vTT which is so slender on its dorsal/ventral axis is not clear. What was the driving depth? How were new units identified following new depths?

---

## [Author Response]

Essential revisions:This is likely the first study that characterized the activity of vTT neurons in behaving animals. The reviewers appreciated the novelty of the study, and they found various observations interesting. However, they raised various issues that need to be addressed before publication of this study in eLife. First, the main conclusion that vTT neurons are "scene cells" warrants more careful evaluations. The concept of "scene cells" is neither well-defined nor sufficiently supported by the presented data. For instance, reviewer #1 points out that a "scene cell" suggests that the activity should be relatively stable during each task event, but many neurons appear to exhibit transient activations at transitions between different task events. Reviewer #2 suggests removing the term altogether. We would like to see more concrete definitions and justification of this term. Alternatively, the authors might want to seek for different terms that better summarize the observed activity or not "naming" these cells. Additionally, the reviewers raised a number of technical and interpretational issues as described below. Although addressing these issues likely do not require additional experiments, we would like the authors to address these issues in a revised manuscript.

We are most grateful to you and the reviewers for careful reading and constructive suggestions, which have greatly helped us improve our paper. Exactly in line with the comments, the entire manuscript has been thoroughly rewritten and reconstructed.

The major issues pointed out by the reviewers are the interpretation of activities of vTT cells. First, as they pointed out, we surely do not have sufficient data to define vTT cells as “scene cells”. Therefore, we have decided to exclude the term in the revised paper. Second, although we hypothesized that the characteristic activities of the vTT cells are generated by the top-down inputs from medial prefrontal cortex, we have never directly demonstrated it in the experiment. We think it is an important hypothesis, but we had put too much emphasis on it in our previous manuscript. Therefore, we put less emphasis on it and have excluded the discussion about coincidence detection in the vTT cells. We have rewritten the Discussion section extensively and reconstructed it to be more result-based in the revised manuscript.

In addition, we have performed reanalysis and added new results to the figures and figure supplements, with reference to the reviewers' comments.

Reviewer #1:[…]Essential revisions:1) The authors measured auROC values and described these frequently as "normalized firing rates" (e.g. Figure 1D, E etc). The auROC is a measure of the discriminability of two distributions and not of response strength. If the authors want to talk about the size and strength of the responses then they should recalculate their data using either raw spike rates, or z-scores if they want to normalize across cells. If they want to simply make the point that response distributions are different then auROC analyses are fine.

Thank you for your pointing it out. We agree with this comment that the auROC is a measure of discriminability of two distributions and not of response strength. As we would like to describe that the response distributions are different, we have revised the description regarding the term “normalized firing rates” to “auROC values” as described below.

Figure 1D, 1E, Figure 2B, Figure 2—figure supplement 3, Figure 3C, and Figure 3—figure supplement 4: We have changed the term “normalized firing rates” to “auROC values”.

The temporal resolution of the analyses is pretty course. If the authors use smaller time bins for the physiology as well as the behavioral analyses they may uncover additional dynamics within each epoch. (For the go/no-go assays they use 20 ms bins and a 60 ms smoothing kernel; for the eating/no-eating assay they use 500 ms binning and 500 ms smoothing.) There are a number of reasons why this may be valuable:

We have reanalyzed the data to use smaller time bins (20 ms bin width for the odor-guided go/no-go task; 50 and 100 ms bin widths for the odor-guided eating/no-eating task). The new results have been added to the figures in Figure 2―figure supplement 2 and Figure 3—figure supplement 3. However, these results are similar to the previous ones and additional dynamics within each epoch were not observed for both tasks.

Figure 2—figure supplement 2 and Figure 3—figure supplement 3: We have added the new results to the figures.

2) First, the authors want to convince us that vTT encodes behavioral scenes, and Figure 4A indicates a strong correlation between the behavioral and neural activity durations. However, Figure 4B indicates that a very large fraction of the recorded cells actually fire at the transitions between epochs. If this is the case, then can we really say that cells in vTT encode behavioral scenes?

We first classified vTT cells based on their firing patterns into behavioral epochs that were observed stereotypically in the process of tasks. We then exhibited the correlation between neural activity durations and behavioral epoch durations (Figure 4A). However, as the reviewer pointed out, some cells showed transient phasic activities at the transition between epochs. In addition, we also exhibited that individual vTT neurons had unique tuning characteristics during development of the tasks, demonstrated by the tuning peak and tuning duration (Figure 4B). Based on these results, we hypothesized that individual vTT cells tuned to distinct behavioral contexts, that is, distinct task-elements occurred in relation to the flow of tasks. We also assumed that these activities were modulated not only by external contextual inputs, but also by internal information such as motivation, memory retrieval, prediction, decision-making, and so on. Then, we named such externally and internally modulated cells as “scene cells”. However, the definition of scene cells in the previous manuscript was inadequate and there were not enough data to support the term. We have therefore excluded the term in the revised manuscript.

3) Second, as very little is known about vTT additional insight might be gained by using a more fine-grained analysis. For example, in Figure 4A, the authors show a strong correlation between median behavioral duration and median firing activity duration. However, the eating epoch, which drives the strong correlation in 4A, has a bimodal distribution of neural activity durations, with many cells having brief responses (< 1 second) and other cells having responses that last for most of the eating epoch. I suspect that there could be multiple classes of neurons (e.g. a transient food odor population and a sustained eating population) embedded within their relatively small number of existing classes.

We thank the reviewer for pointing out the above important issue. We have analyzed the bimodal distribution in the eating epoch. We have demonstrated that the tonic responsive cells tended to be uniformly distributed and the phasic responsive cells were mainly distributed just before dish removal (Figure 4***―***figure supplement 4). It is presumed that the majority of the phasic responsive cells in the transition between approach and eating epochs were contained in the approach epoch.

Figure 4—figure supplement 4: We have added the new result to the supplemental figure.

Subsection “Context-dependent neural activity in the vTT”: We have revised the Discussion section.

4) In the Discussion section, the authors propose that vTT neurons may be coincidence detectors for sensory and top-down inputs. Their tracing studies, which are consistent with older studies, indeed indicate that vTT receives mPFC input, which could be the source of top-down input. However, there are at least two reasons why their results are inconsistent with this model. First, only 25% of their cells were odor responsive yet most of the other cells responded at one or another of the behavioral epochs, even when there was ostensibly no (strong) olfactory input; second, the odor responses to two different odors were essentially identical, Both of these are difficult to reconcile with a system that is strongly driven, or even contingent on direct (from bulb) olfactory input, and instead suggest a circuit that is predominantly driven by top-down input and has surprisingly little (if any) bottom-up sensory drive. For example, do the authors think that vTT neurons would respond similarly or differently to an auditory go/no-go task?

We speculate that the mechanism of coincidence detection, even if unable to distinguish between two cue odors, may work to determine which behavioral states the olfactory sensory inputs enter. Moreover, strong top-down inputs may play a role in inducing plastic change in the weak sensory inputs. However, we have not performed experiments to verify these hypotheses. Before anything, it is important to verify whether top-down inputs from medial prefrontal cortex actually modulate vTT firing. Therefore, we have decided to exclude the discussion for the coincidence detection in revised manuscript. Although we also are interested in the reviewers last question, that is, whether vTT cells respond to other sensory modalities, e.g., auditory one, we have not discussed the issue in the present manuscript.

We have excluded the discussion for the coincidence detection.

Reviewer #2:[…]Essential revisions:1) The authors include data on the chemical identity of vTT neurons (not those recorded from necessarily) and also their connectivity which is tangential to the main story and does not directly support or clarify any pressing matters needed to support major conclusions. We appreciate what the authors are trying to do with these data (helping readers know where input arrives from, and what the output may influence), but these data are not worthy of being included in this manuscript and do not add to the main message in a meaningful way. The main story is the unit data – which is clear, strong, and on its own makes an excellent paper.

Thank you for your positive comment of this paper about unit recording from the vTT. We provided the first known recording of neuronal activity in the vTT during performing odor-guided tasks. Although there have been several reports about anatomical basis of the vTT, they didn’t clarify it quantitatively. Our data provides the first quantitative anatomical information of the vTT. Although the anatomical data in this study does not directly support the major conclusions from the given electrophysiological data, as you commented, we believe that the presentation of both anatomical and electrophysiological data enables readers to understand the interpretations and working hypotheses in this paper. Therefore, we would like to keep the anatomical data in this paper.

2) There is pervasive speculation in this paper and also instances of hypotheses being foreshadowed as to be tested, but never acted upon.Regarding the latter, the authors state the hypothesis in the Introduction that higher order top-down inputs are generating changes in the vTT – yet this hypothesis is never tested in this paper. This is never tested and the anatomy in question above (point 1) does not test that hypothesis neither.We are all for some appropriate scholarly speculation in the Discussion section, but in many cases it reads as if the authors are possibly writing about data they have yet to acquire or just did not include. The speculation in Discussion section that learned odor information about certain contexts is handled "mainly" by the vTT is not warranted by these data which only investigated [albeit compellingly and carefully] vTT activity. At a simplistic level, it is not intuitive that such a small brain region would exert major behavioral influence in contrast to other regions like PCX or OT which also encode learned odor information and have vast connectivity with downstream structures important with affect and behavior. Also, there are many mentions of top-down inputs to vTT being important mediators of the task modulation shown by the authors. Yet this is never shown and are not necessary for the main story of the paper. These findings are interesting in themselves and worthy of publication without having to try to make some circuitry story embedded within. The authors need to seriously restrict speculation about top-down input and also again, remove hypotheses about top-down inputs being integral for the unit activity they report. Focusing the discussion on why the results at hand matter and how they advance the field would be appreciated. Including more discussion on how these results compare/contrast with behavioral modulation of PCX, OT, and OB units would be helpful to appreciate if and how vTT is unique.

As the reviewer pointed out, we have not directly verified the hypothesis of top-down modulation of the vTT. As described in the response to reviewer #1, we have excluded the discussion about coincidence detection and put less emphasis on that point throughout the manuscript. We have then increased the result-based discussion.

Subsection “Context-dependent neural activity in the vTT”: We have revised the Discussion section.

3) The authors define the modulated vTT cells as "scene" cells: "We found that the firing of individual vTT neurons during odor-guided goal-directed behaviors was highly tuned to distinct behavioral "scenes" (i.e., distinct task-elements that occurred in relation to the flow of goal-directed tasks, with each task-element occurring in a specific behavioral context). This is an attractive term, sure, but it is arguable then that almost all single units monitored in the olfactory system, or many other brain regions for that matter, could also be called "scene" cells and thus packaging them with this name is not optimal. For instance, in 1980 Karpov recorded mitral and tufted cells from behaving rabbits as they approached boxes containing odor which they were motivated for. The authors reported spiking upon approach, upon odor, and upon food intake. These results [while certainly not with the rigor in analyses as the authors used in the present manuscript] indicate that mitral cells encode "distinct task-elements that occurred in relation to the flow of goal-directed tasks, with each task-element occurring in a specific behavioral context" – also fitting the authors definition of a "scene cell". We could say the same if we compare and contrast the work herein to many other awake unit recording papers whether in the context of odors or not, where the authors report, of all task modulated units, some are modulated upon anticipation, instrumental responding, stimulus delivery, reward seeking, reward acquisition, etc. There is a possibly latent reason in every one of these papers that authors do not try to name the units with a descriptor – ultimately the descriptor would become useless in the field. This is unlike descriptors bestowed upon other cell types, like "head direction" cells – in these cases one must do many manipulations while recording from the same cell to prove head-direction encoding. Fortunately, this is an easy fix for the authors to remove "scene" throughout the paper. Including a succinct discussion on this concept however is very welcome.

As described in the responses to the editor and reviewer #1, we have decided to exclude the term in the manuscript.

4) It seems the use of the decoding is not fully taken advantage of by the authors. The decoding should be compared to a shuffled baseline, or statistically validated somehow, to see at what point the classification accuracy is significantly greater than chance…. in this paper the authors use it only to say that when trained on all units, the classifier increases in accuracy before odor port exit (so it tracks with the behavior and not the odor itself). But wouldn't it be more compelling to see how classification accuracy differs across the populations they identified? This would strengthen the conclusions that these different populations of modulated neurons are handling information differently. This we will leave at the will of the authors though since, while likely informative, it may delay publication.

We have compared the decoding accuracy with the shuffled baseline and have revised Figure 5B.

The different response patterns of vTT cells between go and no-go trials despite the same nose-poking behaviors were shown in each classified cell group. Because this data was not shown in the previous manuscript, we have added it to the revised one (Figure 5A). Note that populations of these cells were small. To verify whether these firing-rate differences had any contextual information at population levels, we performed the decoding analysis.

5) Details on adjustment of tetrode locations between recording sessions is needed to ensure redundant sampling of units did not occur. The authors state some mice were recorded from for 18 sessions (in table), but if the tetrodes were driven enough to rule out resampling the same units on subsequent days, but how 18 movements can occur in the vTT which is so slender on its dorsal/ventral axis is not clear. What was the driving depth? How were new units identified following new depths?

After every recording session per day, we lowed the tetrodes by 20, 40, or 75 μm more by turning a screw of the microdrive to obtain new units. Electrode tracks were identified by observing the fluorescence of DiI. Recording positions were identified by the driving distances of tetrodes calculated by numbers of screw turning, the positions of electrode tips electrically lesioned after experiments, and the electrode tracks described above. Based on these anatomically identified recording sites, we confirmed that the recording positions were in the vTT.

In order to ascertain that the cells recorded in different days were completely different, we used four waveforms of spikes from four electrode of a tetrode and/or background basic frequency of each unit before the behavioral tasks on successive recording days.

In addition, to exactly exclude the possibility that the same cells were recorded from different tetrodes of the same tetrode bundle, we added an analysis to the revised manuscript. We calculated cross-correlation histograms with 1 msec time bins for all pairs of cells that were recorded from different tetrodes of the same tetrode bundle. If the frequency at 0 msec was 10 × larger than the mean frequency during the baseline and their PETHs had similar shapes, either one of the pair was removed from the database. This method is based on Ito and Doya, (2015). According to the result of the analysis, we have decided to exclude 13 cells in the go/no-go task and 10 cells in the eating/no-eating task.

When we again checked each recording session to see if the electrode driving was appropriately conducted, we noticed that, in several sessions, we had not moved the tetrodes after the previous session. Therefore, we have excluded such inappropriate sessions, resulting in that 5 cells in the go/no-go task and 7 cells in the eating/no-eating task have been excluded from the datasets.

Consequently, in total, we have used 270 cells in the go/no-go task and 374 cells in the eating/no-eating task for the analysis. The reanalysis using these datasets yielded similar results to the previous ones and there was no bias in the distribution of each classified cell (Figure 4―figure supplement 2 and Figure 4―figure supplement 3).